# Probability Flow Solution of the Fokker-Planck Equation

## Abstract

The method of choice for integrating the time-dependent Fokker-Planck equation in high-dimension is to generate samples from the solution via integration of the associated stochastic differential equation. Here, we introduce an alternative scheme based on integrating an *ordinary* differential equation that describes the flow of probability. Acting as a transport map, this equation deterministically pushes samples from the initial density onto samples from the solution at any later time. Unlike integration of the stochastic dynamics, the method has the advantage of giving direct access to quantities that are challenging to estimate from trajectories alone, such as the probability current, the density itself, and its entropy. The probability flow equation depends on the gradient of the logarithm of the solution (its "score"), and so is *a-priori* unknown. To resolve this dependence, we model the score with a deep neural network that is learned on-the-fly by propagating a set of samples according to the instantaneous probability current. We consider several high-dimensional examples from the physics of interacting particle systems to highlight the efficiency and precision of the approach; we find that the method accurately matches analytical solutions computed by hand and moments computed via Monte-Carlo.

## 1 Introduction

The time evolution of many dynamical processes occurring in the natural sciences, engineering, economics, and statistics are naturally described in the language of stochastic differential equations (SDE) (Gardiner, 2009; Oksendal, 2003; Evans, 2012). Typically, one is interested in the probability density function (PDF) of these processes, which describes the probability that the system will occupy a given state at a given time. The density can be obtained as the solution to a Fokker-Planck equation (FPE), which can generically be written as (Risken, 1996; Bass, 2011)

$$\partial_t \rho_t^*(x) = -\nabla \cdot (b_t(x)\rho_t^*(x) - D_t(x)\nabla\rho_t^*(x)), \qquad x \in \Omega \subseteq \mathbb{R}^d, \tag{FPE}$$

where $\rho_t^*(x) \in \mathbb{R}_{\geq 0}$ denotes the value of the density at time $t$, $b_t(x) \in \mathbb{R}^d$ is a vector field known as the drift, and $D_t(x) \in \mathbb{R}^{d \times d}$ is a positive-semidefinite tensor known as the diffusion matrix. (FPE) must be solved for $t \geq 0$ from some initial condition $\rho_{t=0}^*(x) = \rho_0(x)$, but in all but the simplest cases, the solution is not available analytically and can only be approximated via numerical integration.

**High-dimensionality.** For many systems of interest – such as interacting particle systems in statistical physics (Chandler, 1987; Spohn, 2012), stochastic control systems (Kushner et al., 2001), and models in mathematical finance (Oksendal, 2003) – the dimensionality $d$ can be very large. This renders standard numerical methods for partial differential equations inapplicable, which become infeasible for $d$ as small as five or six due to an exponential scaling of the computational complexity with $d$. The standard solution to this problem is a Monte-Carlo approach, whereby the SDE associated with (FPE)

$$dx_t = b_t(x_t)dt + \nabla \cdot D_t(x_t)dt + \sqrt{2}\sigma_t(x_t)dW_t, \tag{1}$$

is evolved via numerical integration to obtain a large number $n$ of trajectories (Kloeden & Platen, 1992). In (1), $\sigma_t(x)$ satisfies $\sigma_t(x)\sigma_t^\mathsf{T}(x) = D_t(x)$ and $W_t$ is a standard Brownian motion on $\mathbb{R}^d$. Assuming that we can draw samples $\{x_0^i\}_{i=1}^n$ from the initial PDF $\rho_0$, simulation of (1) enables the

estimation of expectations via empirical averages

$$\int_\Omega \phi(x)\rho_t^*(x)dx \approx \frac{1}{n}\sum_{i=1}^n \phi(x_t^i), \tag{2}$$

where $\phi : \Omega \to \mathbb{R}$ is an observable of interest. While widely used, this method only provides samples from $\rho_t^*$, and hence other quantities of interest like the value of $\rho_t^*$ itself or the time-dependent differential entropy of the system $H_t = -\int_\Omega \log \rho_t^*(x)\rho_t^*(x)dx$ require sophisticated interpolation methods that typically do not scale well to high-dimension.

**A transport map approach.** Another possibility, building on recent theoretical advances that connect transportation of measures to the Fokker-Planck equation (Jordan et al., 1998), is to recast (FPE) as the transport equation (Villani, 2009; Santambrogio, 2015)

$$\partial_t \rho_t^*(x) = -\nabla \cdot (v_t^*(x)\rho_t^*(x)) \tag{3}$$

where we have defined the velocity field

$$v_t^*(x) = b_t(x) - D_t(x)\nabla \log \rho_t^*(x). \tag{4}$$

This formulation reveals that $\rho_t^*$ can be viewed as the pushforward of $\rho_0$ under the flow map $X_{\tau,t}^*(\cdot)$ of the ordinary differential equation

$$\frac{d}{dt}X_{\tau,t}^*(x) = v_t^*(X_{\tau,t}^*(x)), \qquad X_{\tau,\tau}^*(x) = x, \quad t, \tau \geq 0. \tag{5}$$

Equation (5) is known as the *probability flow equation*, and its solution has the remarkable property that if $x$ is a sample from $\rho_0$, then $X_{0,t}^*(x)$ will be a sample from $\rho_t^*$. Viewing $X_{\tau,t}^* : \Omega \to \Omega$ as a transport map, $\rho_t^* = X_{0,t}^* \sharp \rho_0$ can be evaluated at any position in $\Omega$ via the change of variables formula (Villani, 2009; Santambrogio, 2015)

$$\rho_t^*(x) = \rho_0(X_{t,0}^*(x)) \exp\left(-\int_0^t \nabla \cdot v_\tau^*(X_{t,\tau}^*(x))d\tau\right) \tag{6}$$

where $X_{t,0}^*(x)$ is obtained by solving (5) backward from some given $x$. Importantly, access to the PDF as provided by (6) immediately gives the ability to compute quantities such as the probability current or the entropy; by contrast, this capability is absent when directly simulating the SDE.

**Learning the flow.** The simplicity of the probability flow equation (5) is somewhat deceptive, because the velocity $v_t^*$ depends explicitly on the solution $\rho_t^*$ to the Fokker-Planck equation (FPE). Nevertheless, recent work in generative modeling via score-based diffusion (Song & Ermon, 2020a;b; Song & Kingma, 2021) has shown that it is possible to use deep neural networks to estimate $v_t^*$, or equivalently the so-called *score* $\nabla \log \rho_t^*$ of the solution density. Here, we introduce a variant of score-based diffusion modeling in which the score is learned on-the-fly over samples generated by the probability flow equation itself. The method is self-contained and enables us to bypass simulation of the SDE entirely; moreover, we provide both empirical and theoretical evidence that the resulting self-consistent training procedure offers improved performance when compared to training via samples produced from simulation of the SDE.

## 1.1 CONTRIBUTIONS

Our contributions are both theoretical and computational:

- We provide a bound on the Kullback-Leibler divergence from the estimate $\rho_t$ produced via an approximate velocity field $v_t$ to the target $\rho_t^*$. This bound motivates our approach, and shows that minimizing the discrepancy between the learned score and the score of the push-forward distribution systematically improves the accuracy of $\rho_t$.

- Based on this bound, we introduce two optimization problems that can be used to learn the velocity field (4) in the transport equation (3) so that its solution coincides with that of the Fokker Planck equation (FPE). Due to its similarities with score-based diffusion approaches in generative modeling (SBDM), we call the resulting method *score-based transport modeling* (SBTM).

- We provide specific estimators for quantities that can be computed via SBTM but are not directly available from samples alone, like point-wise evaluation of $\rho_t$ itself, the differential entropy, and the probability current.

- We test SBTM on several examples involving interacting particles that pairwise repel but are kept close by common attraction to a moving trap. In these systems, the FPE is high-dimensional due to the large number of particles, which vary from 5 to 50 in the examples below. Problems of this type frequently appear in the molecular dynamics of externally driven soft matter systems (Frenkel & Smit, 2001; Spohn, 2012). We show that our method can be used to accurately compute the entropy production rate of a system, a quantity of interest in the active matter community (Nardini et al., 2017), as it quantifies the out-of-equilibrium nature of the system's dynamics.

## 1.2 NOTATION AND ASSUMPTIONS.

Throughout, we assume that the stochastic process (1) evolves over a domain $\Omega \subseteq \mathbb{R}^d$ in which it remains at all times $t \geq 0$. We assume that the drift vector $b_t : \Omega \to \mathbb{R}^d$ and the diffusion tensor $D_t : \Omega \to \mathbb{R}^{d \times d}$ are twice-differentiable and bounded in both $x$ and $t$, so that the solution to the SDE (1) is well-defined at all times $t \geq 0$. The symmetric tensor $D_t(x) = D_t^\mathsf{T}(x)$ is assumed to be positive semi-definite for each $(t, x)$, with Cholesky decomposition $D_t(x) = \sigma_t(x)\sigma_t^\mathsf{T}(x)$. We further assume that the initial PDF $\rho_0$ is three-times differentiable, positive everywhere on $\Omega$, and such that $H_0 = -\int_\Omega \log \rho_0(x)\rho_0(x)dx < \infty$. This guarantees that $\rho_t^*$ enjoys the same properties at all times $t > 0$. Finally, we assume that $\log \rho_t^*$ is $K$-smooth globally for $(t, x) \in [0, \infty) \times \Omega$, i.e.

$$\exists K > 0 : \quad \forall (t, x) \in [0, \infty) \times \Omega \quad |\nabla \log \rho_t^*(x) - \nabla \log \rho_t^*(y)| \leq K|x - y|. \tag{7}$$

This technical assumption is needed to guarantee global existence and uniqueness of the solution of the probability flow equation. Throughout, we use the shorthand notation $\dot{y}_t = \frac{d}{dt}y_t$ interchangeably for a time-dependent quantity $y_t$.

## 2 RELATED WORK

**Score matching** Our approach builds directly on the toolbox of score matching originally developed by Hyvärinen (Hyvärinen, 2005; Hyvarinen, 2007; Hyvärinen, 2007; 2008) and more recently extended in the context of diffusion-based generative modeling (Song & Ermon, 2020a;b; Song et al., 2021; De Bortoli et al., 2021; Dockhorn et al., 2022; Mittal et al., 2021). These approaches assume access to training samples from the target distribution (e.g., in the form of examples of natural images). Here, we bypass this need and use the probability flow equation to obtain the samples needed to learn an approximation of the score. Lu et al. (2022) recently showed that using the transport equation (TE) with a velocity field learned via SBDM can lead to inaccuracies in the likelihood unless higher-order score terms are well-approximated. Proposition 1 shows that the self-consistent approach used in SBTM solves these issues and ensures a systematic approximation of the target $\rho_t^*$.

**Density estimation and Bayesian inference** Our method shares commonalities with transport map-based approaches (Marzouk et al., 2016) for density estimation and variational inference (Zhang et al., 2019; Blei et al., 2017) such as normalizing flows (Tabak & Vanden-Eijnden, 2010; Tabak & Turner, 2013; Rezende & Mohamed, 2016; Huang et al., 2021; Papamakarios et al., 2021; Kobyzev et al., 2021). Moreover, because expectations are approximated over a set of samples according to (2), the method also inherits elements of classical "particle-based" approaches for density estimation such as Markov chain Monte Carlo (Robert & Casella, 2004) and sequential Monte Carlo (Dai et al., 2020; Del Moral et al., 2006).

Our approach is also reminiscent of a recent line of work in Bayesian inference that aims to combine the strengths of particle methods with those of variational approximations (Dai et al., 2016; Saeedi et al., 2017). In particular, the method we propose bears some similarity with Stein variational gradient descent (SVGD) (Liu, 2017; Liu & Wang, 2018; 2019) (see also (Lu et al., 2018; Li et al., 2020)), in that both methods approximate the target distribution via *deterministic* propagation of a set of samples. The key differences are that (i) our method learns the map used to propagate the samples, while the map in SVGD corresponds to optimization of the kernelized Stein discrepancy, and (ii) the methods have distinct goals, as we are interested in capturing the dynamical evolution of $\rho_t^*$ rather than sampling at equilibrium.

**Approaches for solving the FPE** Most closely connected to our paper are the works by Maoutsa et al. (2020) and Shen et al. (2022), who similarly propose to bypass the SDE through use of the probability flow equation, building on earlier work by Degond & Mustieles (1990) and Russo (1990). The critical

differences between Maoutsa et al. (2020) and our approach are that they perform estimation over a linear space or a reproducing kernel Hilbert space rather than over the significantly richer class of neural networks, and that they train using the original score matching loss of Hyvärinen (2005), while the use of neural networks requires the introduction of regularized variants. Because of this, Maoutsa et al. (2020) studies systems of dimension less than or equal to five; in contrast, we study systems with dimensionality as high as 100.

Concurrently to our work, Shen et al. (2022) proposed a variational problem similar to SBTM. A key difference is that SBTM is not limited to Fokker-Planck equations that can be viewed as a gradient flow in the Wasserstein metric over some energy (i.e., the drift term in the SDE (1) need not be the gradient of a potential), and that it allows for spatially-dependent and rank-deficient diffusion matrices; moreover, our theoretical results are similar but avoid the use of costly Sobolev norms.

**Neural-network solutions to PDEs** Our approach can also be viewed as an alternative to recent neural network-based methods for the solution of partial differential equations (see e.g. E & Yu (2017); Raissi et al. (2019); Han et al. (2018); Sirignano & Spiliopoulos (2018); Bruna et al. (2022)). Unlike these existing approaches, our method is tailored to the solution of the Fokker-Planck equation and guarantees that the solution is a valid probability density. Our approach is fundamentally Lagrangian in nature, which has the advantage that it only involves learning quantities locally at the positions of a set of evolving samples; this is naturally conducive to efficient scaling for high-dimensional systems.

## 3 METHODOLOGY

### 3.1 SCORE-BASED TRANSPORT MODELING

Let $s_t : \Omega \to \mathbb{R}^d$ denote an approximation to the score of the target $\nabla \log \rho_t^*$, and consider the solution $\rho_t : \Omega \to \mathbb{R}_{\geq 0}$ to the transport equation

$$\partial_t \rho_t(x) = -\nabla \cdot (v_t(x)\rho_t(x)) \qquad \text{with} \quad v_t(x) = b_t(x) - D_t(x)s_t(x). \tag{TE}$$

Our goal is to develop a variational principle that may be used to adjust $s_t$ so that $\rho_t$ tracks $\rho_t^*$. Our approach is based on the following inequality, whose proof may be found in Appendix B.1:

**Proposition 1** (Control of the KL divergence). *Assume that the conditions listed in Sec. 1.2 hold. Let $\rho_t$ denote the solution to the transport equation* (TE)*, and let $\rho_t^*$ denote the solution to the Fokker-Planck equation* (FPE)*. Assume that $\rho_{t=0}(x) = \rho_{t=0}^*(x) = \rho_0(x)$ for all $x \in \Omega$. Then*

$$\frac{d}{dt}D_{\mathsf{KL}}(\rho_t \mid \rho_t^*) \leq \frac{1}{2}\int_\Omega |s_t(x) - \nabla \log \rho_t(x)|^2_{D_t(x)}\,\rho_t(x)dx, \tag{8}$$

*where $|\cdot|^2_{D_t(x)} = \langle\cdot, D_t(x)\cdot\rangle$.*

In particular, (8) implies that for any $T \in [0, \infty)$ we have explicit control on the KL divergence

$$D_{\mathsf{KL}}(\rho_T \mid \rho_T^*) \leq \frac{1}{2}\int_0^T \int_\Omega |s_t(x) - \nabla \log \rho_t(x)|^2_{D_t(x)}\,\rho_t(x)dxdt. \tag{9}$$

Remarkably, (9) only depends on the approximate $\rho_t$ and does not include $\rho_t^*$: it states that the accuracy of $\rho_t$ as an approximation of $\rho_t^*$ can be improved by enforcing agreement between $s_t$ and $\nabla \log \rho_t$. This means that we can optimize (9) *without* making use of external data from $\rho_t^*$, which offers a self-consistent objective function to learn the score $s_t$ using (TE) alone.

The primary difficulty with this approach is that $\rho_t$ must be considered as a functional of $s_t$, since the velocity $v_t$ used in (TE) depends on $s_t$. To render the resulting minimization of the right-hand side of (9) practical, we can exploit that (TE) can be solved via the method of characteristics, as summarized in Appendix A. Specifically, if $\dot{X}_t(x) = v_t(X_t(x))$ is the probability flow equation associated with the velocity $v_t$, then $\rho_t = X_t \sharp \rho_0$. This means that the expectation of any function $\phi(x)$ over $\rho_t(x)$ can be expressed as the expectation of $\phi_t(X_t(x))$ over $\rho_0(x)$. Observing that the score of the solution to (TE) along trajectories of the probability flow $\nabla \log \rho_t(X_t(x))$ solves a closed equation leads to the following proposition.

**Proposition 2** (Score-based transport modeling). *Assume that the conditions listed in Sec. 1.2 hold. Define $v_t(x) = b_t(x) - D_t(x)s_t(x)$ and consider*

$$\begin{aligned} \dot{X}_t(x) &= v_t(X_t(x)), & X_0(x) &= x, \\ \dot{G}_t(x) &= -[\nabla v_t(X_t(x))]^\mathsf{T} G_t(x) - \nabla\nabla \cdot v_t(X_t(x)), & G_0(x) &= \nabla \log \rho_0(x). \end{aligned} \tag{10}$$

*Then $\rho_t = X_t \sharp \rho_0$ solves* (TE)*, the equality $G_t(x) = \nabla \log \rho_t(X_t(x))$ holds, and for any $T \in [0, \infty)$*

$$D_{\mathsf{KL}}(X_T \sharp \rho_0 \mid \rho_T^*) \leq \frac{1}{2} \int_0^T \int_\Omega |s_t(X_t(x)) - G_t(x)|^2_{D_t(X_t(x))} \rho_0(x) dx dt. \tag{11}$$

*Moreover, if $s_t^*$ is a minimizer of the constrained optimization problem*

$$\min_s \int_0^T \int_\Omega |s_t(X_t(x)) - G_t(x)|^2_{D_t(X_t(x))} \rho_0(x) dx dt \quad \text{subject to (10)} \tag{SBTM}$$

*then $D_t(x)s_t^*(x) = D_t(x)\nabla \log \rho_t^*(x)$ where $\rho_t^*$ solves the Fokker-Planck equation* (FPE)*. The map $X_t^*$ associated to any minimizer is a transport map from $\rho_0$ to $\rho_t^*$, i.e.*

$$x \sim \rho_0 \qquad \text{implies that} \qquad X_t^*(x) \sim \rho_t^*, \qquad \forall t \in [0, T]. \tag{12}$$

Proposition 2 is proven in Appendix B.3. The result also holds with a standard Euclidean norm replacing the diffusion-weighted norm, in which case the minimizer is unique and is given by $s_t^*(x) = \nabla \log \rho_t^*(x)$. In the special case when the SDE is an Ornstein-Uhlenbeck process, the score and the equations for both $X_t$ and $G_t$ can be written explicitly; they are studied in Appendix C.

In practice, the objective in (SBTM) can be estimated empirically by generating samples from $\rho_0$ and solving the equations for $X_t(x)$ and $G_t(x)$ with $x \sim \rho_0$. The constrained minimization problem (SBTM) can then in principle be solved with gradient-based techniques via the adjoint method. The corresponding equations are written in Appendix B.3, but they involve fourth-order spatial derivatives that are computationally expensive to compute via automatic differentiation. Moreover, each gradient step requires solving a system of ordinary differential equations whose dimensionality is equal to the number of samples used to compute expectations times the dimension of (FPE). Instead, we now develop a sequential procedure that avoids these difficulties entirely.

## 3.2 SEQUENTIAL SCORE-BASED TRANSPORT MODELING

An alternative to the constrained minimization in Proposition 2 is to consider an approach whereby the score $s_t$ is obtained independently at each time to ensure that $D_{\mathsf{KL}}(\rho_t \mid \rho_t^*)$ remains small. This suggests choosing $s_t$ to minimize $\frac{d}{dt} D_{\mathsf{KL}}(\rho_t \mid \rho_t^*)$, which admits a simple closed-form bound, as shown in Proposition 1. While this explicit form can be used directly, an application of Stein's identity recovers an implicit objective analogous to Hyvärinen score-matching that is equivalent to minimizing $\frac{d}{dt} D_{\mathsf{KL}}(\rho_t \mid \rho_t^*)$ but obviates the calculation of $G_t$. Expanding the square in (8) and applying $\int_\Omega s_t(x)^\mathsf{T} \nabla \log \rho_t(x) \, \rho_t(x) dx = -\int_\Omega \nabla \cdot s_t(x) \, \rho_t(x) dx$, we may write

$$\frac{d}{dt} D_{\mathsf{KL}}(\rho_t \mid \rho_t^*) \leq \frac{1}{2} \int_\Omega \left( |s_t(X_t(x))|^2_{D_t(X_t(x))} + 2\nabla \cdot (D_t(X_t(x))s_t(X_t(x))) + |G_t(x)|^2 \right) \rho_0(x) dx.$$

Because $\nabla \log \rho_t(X_t(x)) = G_t(x)$ is independent of $s_t$, we may neglect the corresponding square term during the optimization. This leads to a simple and comparatively less expensive way to build the pushforward $X_t^*$ such that $X_t^* \sharp \rho_0 = \rho_t^*$ sequentially in time, as stated in the following proposition.

**Proposition 3** (Sequential SBTM)**.** *In the same setting as Proposition 2, let $X_t(x)$ solve the first equation in (10) with $v_t(x) = b_t(x) - D_t(x)s_t(x)$. Let $s_t$ be obtained via*

$$\min_{s_t} \int_\Omega \left( |s_t(X_t(x))|^2_{D_t(X_t(x))} + 2\nabla \cdot (D_t(X_t(x))s_t(X_t(x))) \right) \rho_0(x) dx. \tag{SSBTM}$$

*Then, each minimizer $s_t^*$ of* (SSBTM) *satisfies $D_t(x)s_t^*(x) = D_t(x)\nabla \log \rho_t^*(x)$ where $\rho_t^*$ is the solution to* (FPE)*. Moreover, the map $X_t^*$ associated to $s_t^*$ is a transport map from $\rho_0$ to $\rho_t^*$.*

Proposition 3 is proven in Appendix B.4. Critically, (SSBTM) is no longer a constrained optimization problem. Given the current value of $X_t$ at any time $t$, we can obtain $s_t$ via direct minimization of the objective in (SSBTM). Given $s_t$, we may compute the right-hand side of (10) and propagate $X_t$ (and possibly $G_t$) forward in time. The resulting procedure, which alternates between self-consistent score estimation and sample propagation, is presented in Algorithm 1. The output of the method produces a feasible solution for (SBTM) with an *a-posteriori* bound on the loss obtained via integration.

**Algorithm 1** Sequential score-based transport modeling.

---

1: **Input**: An initial time $t_0 \in \mathbb{R}_{\geq 0}$. A set of $n$ samples $\{x_i\}_{i=1}^n$ from $\rho_{t_0}$. A set of $N_T$ timesteps $\{\Delta t_k\}_{k=0}^{N_T-1}$.

2: Initialize sample locations $X_{t_0}^i = x_i$ for $i = 1, \ldots, n$.

3: **for** $k = 0, \ldots, N_t - 1$ **do**

4:      Optimize: $s_{t_k} = \arg\min_s \frac{1}{n} \sum_{i=1}^n \left[ |s(X_{t_k}^i)|_{D_{t_k}(X_{t_k}^i)}^2 + 2\nabla \cdot \left( D_{t_k}(X_{t_k}^i) s(X_{t_k}^i) \right) \right]$.

5:      Propagate samples:
$$X_{t_{k+1}}^i = X_{t_k}^i + \Delta t_k \left( b_{t_k}(X_{t_k}^i) - D_{t_k}(X_{t_k}^i) s_{t_k}(X_{t_k}^i) \right).$$

6:      Set $t_{k+1} = t_k + \Delta t_k$.

7: **Output**: A set of $n$ samples $\{X_{t_k}^i\}_{i=1}^n$ from $\rho_{t_k}$ and the score $\{s_{t_k}(X_{t_k}^i)\}_{i=1}^n$ for all $\{t_k\}_{k=0}^{N_T}$.

---

**Practical considerations** To avoid computation of the divergence $\nabla \cdot (D_t(X_t(x)) s_t(X_t(x)))$, which is often costly for neural networks, we can use the denoising score matching loss function introduced by Vincent (2011), which we discuss in Appendix B.6. Empirically, we find that the use of either the denoising objective or explicit derivative regularization is necessary for stable training.

**Why not use the SDE?** An alternative to the sequential procedure outlined here would be to generate samples from the target $\rho_t^*$ via simulation of the associated SDE, and to approximate the score $\nabla \log \rho_t^*$ via minimization of the loss $\int_0^T \int_\Omega (|s_t(x)|^2 + 2\nabla \cdot s_t(x)) \rho_t^*(x) dx dt$, similar to SBDM. As shown in Appendix B.5 neither $D_{\mathsf{KL}}(\rho_t \mid \rho_t^*)$ nor $D_{\mathsf{KL}}(\rho_t^* \mid \rho_t)$ are controlled when using this procedure, where $\rho_t = X_t \sharp \rho_0$ is the density of the probability flow equation. Empirically, we find in the numerical experiments that this approach is significantly less numerically stable than sequential SBTM. In particular, we could not estimate the entropy using the score learned from the SDE.

**SBTM vs. Sequential SBTM** Given the simplicity of the optimization problem (SSBTM), one may wonder if (SBTM) is useful in practice, or if it is simply a stepping stone to arrive at (SSBTM). The primary difference is that (SBTM) offers global control on the discrepancy between $s_t$ and $\nabla \log \rho_t$ over $t \in [0, T]$ that unavoidably arises in practice due to learning and time-discretization errors. By contrast, because (SSBTM) proceeds sequentially, these errors could accumulate over time in a way that is harder to control. In the numerical examples below, we took the timestep $\Delta t$ sufficiently small, and the number of samples $n$ sufficiently large, that we did not observe any accumulation of error. Nevertheless, (SBTM) may allow for more accurate approximation, because the loss is exactly minimized at zero and high-order derivatives of $s_t$ must be controlled through the calculation of $\dot{G}_t$.

## 4 NUMERICAL EXPERIMENTS

In the following, we study two high dimensional examples from the physics of interacting particle systems, where the spatial variable of the Fokker-Planck equation (FPE) can be written as $x = \left( x^{(1)}, x^{(2)}, \ldots, x^{(N)} \right)^\mathsf{T}$ with each $x^{(i)} \in \mathbb{R}^{\bar{d}}$. Here, $\bar{d}$ describes a lower-dimensional ambient space, e.g. $\bar{d} = 2$, so that the dimensionality of the Fokker-Planck equation $d = N\bar{d}$ will be high if the number of particles $N$ is even moderate. The still figures shown below do not do full justice to the complexity of the particle dynamics, and we encourage the reader to view the movies available here. With a timestep $\Delta t = 10^{-3}$, a horizon $T = 10$, and a fixed $nN\bar{d} = 10^5$, we find that the sequential SBTM procedure takes around two hours for each simulation on a single NVIDIA RTX8000 GPU.

### 4.1 HARMONICALLY INTERACTING PARTICLES IN A HARMONIC TRAP

**Setup.** Here we study a problem that admits a tractable analytical solution for direct comparison. We consider $N$ two-dimensional particles ($\bar{d} = 2$) that repel according to a harmonic interaction but experience harmonic abut experience ttraction towards a moving trap $\beta_t \in \mathbb{R}^2$. The motion of the particles is governed by the stochastic dynamics

$$dX_t^{(i)} = (\beta_t - X_t^{(i)})dt + \alpha\left(X_t^{(i)} - \frac{1}{N}\sum_{j=1}^N X_t^{(j)}\right)dt + \sqrt{2D}\,dW_t, \quad i = 1, \ldots, N \quad (13)$$

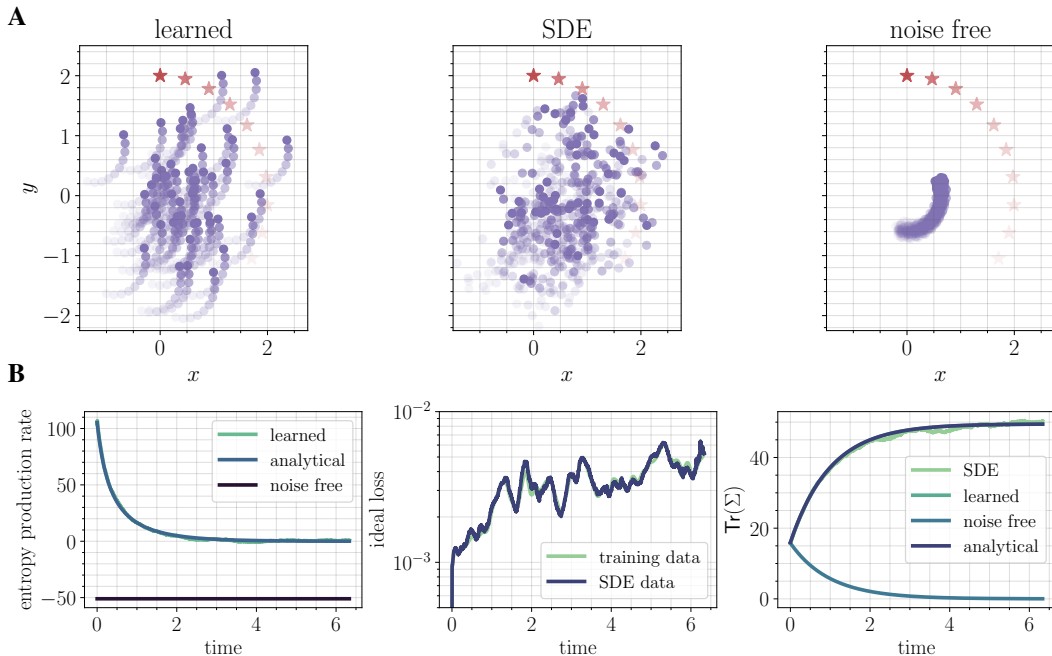

Figure 1: *A system of $N = 50$ particles in a harmonic trap with a harmonic interaction:* (A) A single sample trajectory. The mean of the trap $\beta_t$ is shown with a red star, while past positions of the particles are indicated by a fading trajectory. The noise-free system (right) is too concentrated, and fails to capture the variance of the stochastic dynamics (center). The learned system (left) accurately captures the variance, and in addition generates physically interpretable trajectories for the particles. (B) Quantitative comparison to the analytical solution. The learned solution matches the entropy production rate, score, and covariance well. Movie can be found here.

where $\alpha \in (0,1)$ is a fixed coefficient that sets the magnitude of the repulsion. The dynamics (13) is an Ornstein-Uhlenbeck process in the extended variable $x \in \mathbb{R}^{\bar{d}N}$ with block components $x^{(i)}$. Assuming a Gaussian initial condition, the solution to the Fokker-Planck equation associated with (13) is a Gaussian for all time and hence can be characterized entirely by its mean $m_t$ and covariance $C_t$. These can be obtained analytically (Appendices C and D), which facilitates a quantitative comparison to the learned model. The differential entropy $S_t$ is given by (see Appendix D)

$$H_t = \tfrac{1}{2}\bar{d}N\left(\log(2\pi) + 1\right) + \tfrac{1}{2}\log \det C_t \tag{14}$$

In the experiments, we take $\beta_t = a(\cos \pi\omega t, \sin \pi\omega t)^\mathsf{T}$ with $a = 2$, $\omega = 1$, $D = 0.25$, $\alpha = 0.5$, and $N = 50$, giving rise to a 100-dimensional Fokker-Planck equation. The particles are initialized from an isotropic Gaussian with mean $\beta_0$ (the initial trap position) and variance $\sigma_0^2 = 0.25$.

**Network architecture.** We take $s_t(x) = -\nabla U_{\theta_t}(x)$, where the potential $U_{\theta_t}(\cdot)$ is given as a sum of one- and two-particle terms

$$U_{\theta_t}\left(x^{(1)}, \ldots, x^{(N)}\right) = \sum_{i=1}^{N} U_{\theta_t,1}\left(x^{(i)}\right) + \frac{1}{N}\sum_{\substack{i,j=1 \\ i \neq j}}^{N} U_{\theta_t,2}\left(x^{(i)}, x^{(j)}\right). \tag{15}$$

To ensure permutation symmetry amongst the particles, we require that $U_{\theta_t,2}(x,y) = U_{\theta_t,2}(y,x)$ for each $x, y \in \mathbb{R}^{\bar{d}}$. Modeling at the level of the potential introduces an additional gradient into the loss function, but makes it simple to enforce permutation symmetry; moreover, by writing the potential as a sum of one- and two-particle terms, the dimensionality of the function estimation problem is reduced. As motivation for this choice of architecture, we show in Appendix D.1 that the class of scores representable by (15) contains the analytical score for the harmonic problem considered in this section. To obtain the parameters $\theta_{t_k + \Delta t_k}$, we perform a warm start and initialize from $\theta_{t_k}$, which

reduces the number of optimization steps that need to be performed at each iteration. All networks are taken to be multi-layer perceptrons with the `swish` activation function (Ramachandran et al., 2017); further details on the architectures used can be found in Appendix D.

**Quantitative comparison.** For a quantitative comparison between the learned model and the exact solution, we study the empirical covariance $\Sigma$ over the samples and the entropy production rate $\frac{dS_t}{dt}$. Because an analytical solution is available for this system, we may also compute the target $\nabla \log \rho_t(x) = -C_t^{-1}(x - m_t)$ and measure the goodness of fit via the relative Fisher divergence

$$\frac{\int_\Omega |s_t(x) - \nabla \log \rho_t(x)|^2 \bar{\rho}(x) dx}{\int_\Omega |\nabla \log \rho_t(x)|^2 \bar{\rho}(x) dx}. \tag{16}$$

In Equation (16), $\bar{\rho}$ can be taken to be equal to the current particle estimate of $\rho_t$ (the training data), or estimated using samples from the stochastic differential equation (the SDE data).

**Results.** The representation of the dynamics (13) in terms of the flow of probability leads to an intuitive deterministic motion that accurately captures the statistics of the underlying stochastic process. Snapshots of particle trajectories from the learned probability flow (5), the SDE (13), and the noise-free equation obtained by setting $D = 0$ in (13) are shown in Figure 1A.

Results for this quantitative comparison are shown in Figure 1B. The learned model accurately predicts the entropy production rate of the system and minimizes the relative metric (16) to the order of $10^{-2}$. The noise-free system incorrectly predicts a constant and negative entropy production rate, while the SDE cannot make a prediction for the entropy production rate. In addition, the learned model accurately predicts the high-dimensional covariance $\Sigma$ of the system (curves lie directly on top of the analytical result, trace shown for simplicity). The SDE also captures the covariance, but exhibits more fluctuations in the estimate; the noise-free system incorrectly estimates all covariance components as converging to zero.

## 4.2 SOFT SPHERES IN AN ANHARMONIC TRAP

**Setup.** Here, we consider a system of $N = 5$ particles in an *anharmonic* trap in dimension $\bar{d} = 2$ that exhibit soft-sphere repulsion. This system gives rise to a 10-dimensional (FPE), a dimensionality that is significantly too high for standard PDE solvers. The stochastic dynamics is given by

$$dX_t^{(i)} = 4B(\beta_t - X_t^{(i)})|X_t^{(i)} - \beta_t|^2 dt$$
$$+ \frac{A}{Nr^2} \sum_{j=1}^N (X_t^{(i)} - X_t^{(j)}) \exp\left(-\frac{|X_t^{(i)} - X_t^{(j)}|^2}{2r^2}\right) dt + \sqrt{2D}\, dW_t, \quad i = 1, \ldots, N,$$

where $\beta_t$ again represents a moving trap, $A > 0$ sets the strength of the repulsion between the spheres, $r$ sets their size, and $B > 0$ sets the strength of the trap. We set $\beta(t) = a(\cos \pi \omega t, \sin \pi \omega t)^\mathsf{T}$ or $\beta(t) = a(\cos \pi \omega t, 0)^\mathsf{T}$ with $a = 2, \omega = 1, D = 0.25, A = 10$, and $r = 0.5$. We fix $B = D/R^2$ with $R = \sqrt{\gamma N}r$ and $\gamma = 5.0$. This ensures that the trap scales with the number of particles and that they have sufficient room in the trap to generate a complex dynamics. The circular case converges to a distribution $\rho_t^* = \rho^* \circ Q_t$ that can be described as a fixed distribution $\rho^*$ composed with a time-dependent rotation $Q_t$, and hence the entropy production rate should converge to zero. The linear case does not exhibit such convergence, and the entropy production rate should oscillate around zero as the particles are repeatedly pushed and pulled by the trap. We make use of the same network architecture as in Sec. 4.1.

**Results.** Similar to Section 4.1, an example trajectory from the learned system, the SDE (17) and the noise-free system obtained by setting $D = 0$ in (17) are shown in Figure 2A in the circular case. The learned particle trajectories exhibit an intuitive circular motion with increased disorder relative to the noise-free system that accurately captures the statistics of the stochastic dynamics. Numerical estimates of a single component of the covariance and of the entropy production rate are shown in Figure 2B/C, with all moments shown in Appendix D.2. The learned and SDE systems accurately capture the covariance, while the noise-free system underestimates the covariance in both the linear and the circular case. The prediction of the entropy production rate from Algorithm 1 is reasonable in both cases, exhibiting convergence to zero and oscillation around zero as expected. In the inset, we show the prediction of the entropy production rate when learning on samples from the SDE; the

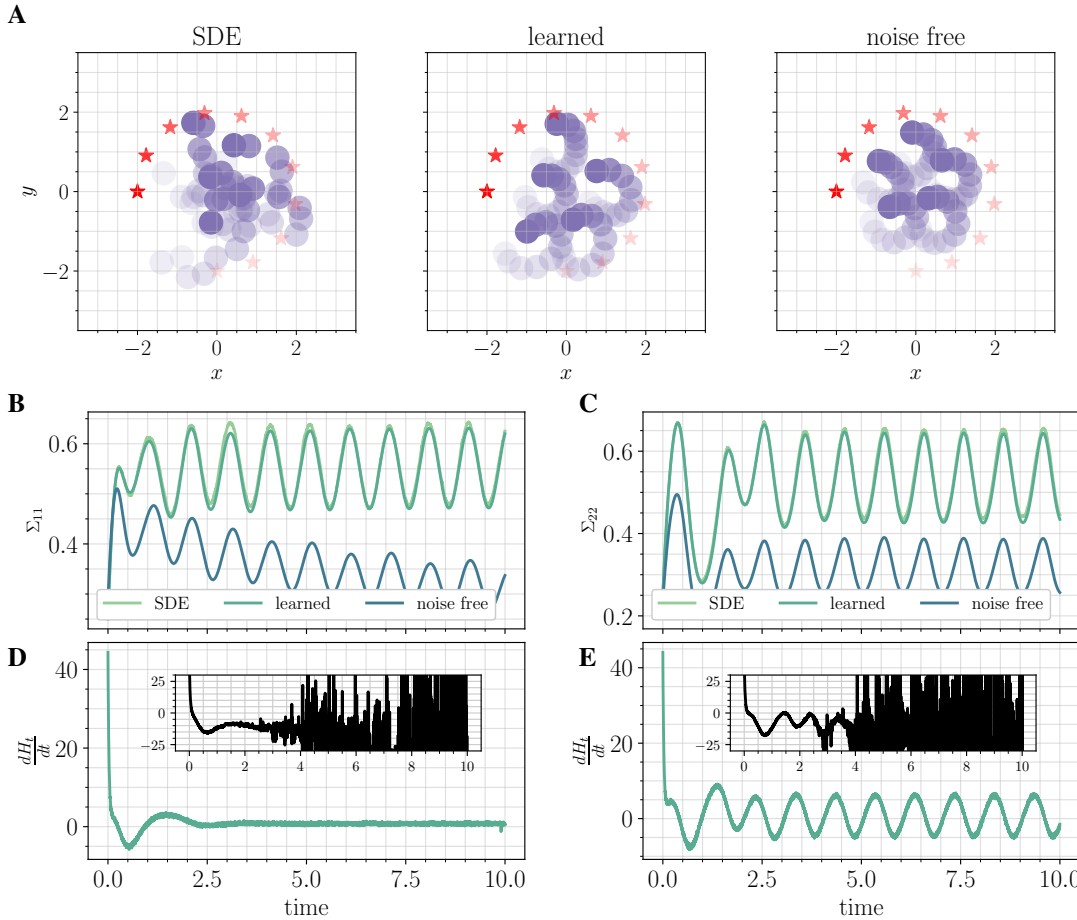

Figure 2: *A system of $N = 5$ soft-spheres in an anharmonic trap:* (A) Example particle trajectories in the case of a rotating trap. Trap position shown with a red star. (B/C) A single component of the covariance of the samples, in the case of a rotating trap (B) and a linearly oscillating trap (C). The system learned system agrees well with the SDE, while the noise-free systems under-predicts the moments. (D/E) Prediction of the entropy production rate for a rotating trap (B) and linearly oscillating trap (C). Main figure depicts prediction from SBTM, while the inset depicts the prediction when learning on SDE samples. SBTM captures the temporal evolution of the entropy production rate, while learning on the SDE is initially offset and later divergent. Movies of the circular and linear motion can be viewed here and here, respectively.

prediction is initially offset, and later becomes divergent. We found that this behavior was generic when training on the SDE, but never observed it when training on self-consistent samples

## 5 OUTLOOK AND CONCLUSIONS

Building on the toolbox of score-based diffusion recently developed for generative modeling, we introduced a related approach – score-based transport modeling (SBTM) – that gives an alternative to simulating the corresponding SDE to solve the Fokker-Planck equation. While SBTM is more costly than integration of the SDE because it involves a learning component, it gives access to quantities that are not directly accessible from the samples given by integrating the SDE, such as pointwise evaluation of the PDF, the probability current, or the entropy. Our numerical examples indicate that SBTM is scalable to systems in high dimension where standard numerical techniques for partial differential equations are inapplicable. The method can be viewed as a deterministic Lagrangian integration method for the Fokker-Planck equation, and our results show that its trajectories are more easily interpretable than the corresponding trajectories of the SDE.

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

## A  Some basic formulas

Here, we derive some results linking the solution of the transport equation (TE) with that of the probability flow equation (5).

### A.1  Probability density and probability current

We begin with a lemma.

**Lemma A.1.** *Let $\rho_t : \Omega \to \mathbb{R}_{\geq 0}$ satisfy the transport equation*

$$\partial_t \rho_t(x) = -\nabla \cdot (v_t(x)\rho_t(x)). \tag{A.1}$$

*Assume that $v_t(x)$ is $C^2$ in both $t$ and $x$ for $t \geq 0$ and globally Lipschitz in $x$. Then, given any $t, t' \geq 0$, the solution of (A.1) satisfies*

$$\rho_t(x) = \rho_{t'}(X_{t,t'}(x)) \exp\left(-\int_{t'}^{t} \nabla \cdot v_\tau(X_{t,\tau}(x))d\tau\right) \tag{A.2}$$

*where $X_{\tau,t}$ is the probability flow solution to (5). In addition, given any test function $\phi : \Omega \to \mathbb{R}$, we have*

$$\int_\Omega \phi(x)\rho_t(x)dx = \int_\Omega \phi(X_{t',t}(x))\rho_{t'}(x)dx. \tag{A.3}$$

In words, Lemma A.1 states that an evaluation of the PDF $\rho_t$ at a given point $x$ may be obtained by evolving the probability flow equation (5) backwards to some earlier time $t'$ to find the point $x'$ that evolves to $x$ at time $t$, assuming that $\rho_{t'}(x')$ is available. In particular, for $t' = 0$, we obtain

$$\rho_t(x) = \rho_0(X_{t,0}(x)) \exp\left(-\int_0^t \nabla \cdot v_\tau(X_{t,\tau}(x))d\tau\right), \tag{A.4}$$

and

$$\int_\Omega \phi(x)\rho_t(x)dx = \int_\Omega \phi(X_{0,t}(x))\rho_0(x)dx. \tag{A.5}$$

Since the probability current is by definition $v_t(x)\rho_t(x)$, using (A.4) to express $\rho_t(x)$ also gives the follwing equation for the current:

$$v_t(x)\rho_t(x) = v_t(x)\rho_0(X_{t,0}(x)) \exp\left(-\int_0^t \nabla \cdot v_\tau(X_{\tau,t}(x))d\tau\right). \tag{A.6}$$

*Proof.* The assumed $C^2$ and globally Lipschitz conditions on $v_t$ guarantee global existence (on $t \geq 0$) and uniqueness of the solution to (5). Differentiating $\rho_t(X_{t',t}(x))$ with respect to $t$ and using (5) and (A.1) we deduce

$$\begin{aligned}
\frac{d}{dt}\rho_t(X_{t',t}(x)) &= \partial_t \rho_t(X_{t',t}(x)) + \frac{d}{dt}X_{t',t}(x) \cdot \nabla\rho_t(X_{t',t}(x)) \\
&= \partial_t \rho_t(X_{t',t}(x)) + v_t(X_{t',t}(x)) \cdot \nabla\rho_t(X_{t',t}(x)) \\
&= -\nabla \cdot v_t(X_{t',t}(x))\, \rho_t(X_{t',t}(x))
\end{aligned} \tag{A.7}$$

Integrating this equation in $t$ from $t = t'$ to $t = t$ gives

$$\rho_t(X_{t',t}(x)) = \rho_{t'}(x) \exp\left(-\int_{t'}^{t} \nabla \cdot v_\tau(X_{t',\tau}(x))d\tau\right) \tag{A.8}$$

Evaluating this expression at $x = X_{t,t'}(x)$ and using the group properties (i) $X_{t',t}(X_{t,t'}(x)) = x$ and (ii) $X_{t',\tau}(X_{t,t'}(x)) = X_{t,\tau}(x)$ gives (A.2). Equation (A.3) can be derived by using (A.2) to express $\rho_t(x)$ in the integral at the left hand-side, changing integration variable $x \to X_{t',t}(x)$ and noting that the factor $\exp\left(-\int_{t'}^{t} \nabla \cdot v_\tau(X_{t,\tau}(x))\right)$ is precisely the Jacobian of this change of variable. The result is the integral at the right hand-side of (A.3). $\qquad\square$

Lemma A.1 also holds locally in time for any $v_t(x)$ that is $C^2$ in both $t$ and $x$. In particular, it holds locally if we set $s_t(x) = \nabla \log \rho_t(x)$ and if we assume that $\rho_0(x)$ is (i) positive everywhere on $\Omega$ and (ii) $C^3$ in $x$. In this case, (A.1) is the Fokker-Planck equation (FPE) and (A.2) holds for the solution to that equation.

## A.2 Calculation of the differential entropy

We now consider computation of the differential entropy, and state a similar result.

**Lemma A.2.** *Assume that $\rho_0 : \Omega \to \mathbb{R}_{\geq 0}$ is positive everywhere on $\Omega$ and $C^3$ in its argument. Let $\rho_t : \Omega \to \mathbb{R}_{\geq 0}$ denote the solution to the Fokker Planck equation* (FPE) *(or equivalently, to the transport equation* (A.1) *with $s_t(x) = \nabla \log \rho_t(x)$ in the definition of $v_t(x)$). Then the differential entropy $s_t = -\int_\Omega \log \rho_t(x)\, \rho_t(x) dx$ can expressed as*

$$H_t = -\int_\Omega \log \rho_t(X_{0,t}(x))\, \rho_0(x) dx = H_0 + \int_0^t \int_\Omega \nabla \cdot v_\tau(X_{0,\tau}(x))\rho_0(x)dx d\tau \qquad \text{(A.9)}$$

*or*

$$H_t = H_0 - \int_0^t \int_\Omega s_\tau(X_{0,\tau}(x)) \cdot v_\tau(X_{0,\tau}(x))\rho_0(x)dx d\tau \qquad \text{(A.10)}$$

*Proof.* We first derive (A.9). Observe that applying (A.5) with $\phi = \log \rho_t$ leads to the first equality. The second can then be deduced from (A.4). To derive (A.10), notice that from (A.1),

$$\begin{aligned}
\frac{d}{dt} H_t &= \int_\Omega \log \rho_t(x) \nabla \cdot (v_t(x)\rho_t(x))\, dx, \\
&= -\int_\Omega \nabla \log \rho_t(x) \cdot v_t(x)\rho_t(x)dx, \\
&= -\int_\Omega s_t(x) \cdot v_t(x)\rho_t(x)dx
\end{aligned} \qquad \text{(A.11)}$$

Above, we used integration by parts to obtain the second equality and $s_t = \nabla \log \rho_t$ to get the third. Now, using (A.5) with $\phi = s_t \cdot v_t$ integrating the result gives (A.10). $\qquad \square$

## A.3 Resampling of $\rho_t$ at any time $t$

If the score $s_t \approx \nabla \log \rho_t$ is known to sufficient accuracy, $\rho_t$ can be resampled at any time $t$ using the dynamics

$$dX_\tau = s_t(X_\tau)d\tau + dW_\tau. \qquad \text{(A.12)}$$

In (A.12), $\tau$ is an artificial time used for sampling that is distinct from the physical time in (1). For $s_t = \nabla \log \rho_t$, the equilibrium distribution of (A.12) is exactly $\rho_t$. In practice, $s_t$ will be imperfect and will have an error that increases away from the samples used to learn it; as a result, (A.12) should be used near samples for a fixed amount of time to avoid the introduction of additional errors.

# B Further details on Score-Based Transport Modeling

## B.1 Bounding the KL divergence

Let us restate Proposition 1 for convenience:

**Proposition 1** (Control of the KL divergence)**.** *Assume that the conditions listed in Sec. 1.2 hold. Let $\rho_t$ denote the solution to the transport equation* (TE)*, and let $\rho_t^*$ denote the solution to the Fokker-Planck equation* (FPE)*. Assume that $\rho_{t=0}(x) = \rho_{t=0}^*(x) = \rho_0(x)$ for all $x \in \Omega$. Then*

$$\frac{d}{dt} D_{\mathsf{KL}}(\rho_t \mid \rho_t^*) \leq \frac{1}{2} \int_\Omega |s_t(x) - \nabla \log \rho_t(x)|_{D_t(x)}^2\, \rho_t(x)dx, \qquad \text{(8)}$$

*where $|\cdot|_{D_t(x)}^2 = \langle \cdot, D_t(x) \cdot \rangle$.*

*Proof.* By assumption, $\rho_t$ solves (TE) and $\rho_t^*$ solves (FPE). Denote by $v_t(x) = b_t(x) - D_t(x)s_t(x)$ and $v_t^*(x) = b_t(x) - D_t(x)s_t^*(x)$ with $s_t^*(x) = \nabla \log \rho_t^*(x)$. Then, we have

$$
\begin{aligned}
\frac{d}{dt} D_{\mathsf{KL}}(\rho_t \mid \rho_t^*) &= \frac{d}{dt} \int_\Omega \log \left( \frac{\rho_t(x)}{\rho_t^*(x)} \right) \rho_t(x)dx, \\
&= -\int_\Omega \frac{\rho_t(x)}{\rho_t^*(x)} \partial_t \rho_t^*(x)dx + \int_\Omega \log \left( \frac{\rho_t(x)}{\rho_t^*(x)} \right) \partial_t \rho_t(x)dx, \\
&= -\int_\Omega v_t^*(x) \cdot \nabla \left( \frac{\rho_t(x)}{\rho_t^*(x)} \right) \rho_t^*(x)dx + \int_\Omega v_t(x) \cdot \nabla \log \left( \frac{\rho_t(x)}{\rho_t^*(x)} \right) \rho_t(x)dx, \\
&= -\int_\Omega \left( v_t^*(x) - v_t(x) \right) \cdot \left( \nabla \log \rho_t(x) - \nabla \log \rho_t^*(x) \right) \rho_t(x)dx, \\
&= \int_\Omega \left( s_t^*(x) - s_t(x) \right) \cdot D_t(x) \left( \nabla \log \rho_t(x) - s_t^*(x) \right) \rho_t(x)dx.
\end{aligned}
$$

Above, we used integration by parts to obtain the third equality. Now, dropping function arguments for simplicity of notation, we have that

$$
\begin{aligned}
|\nabla \log \rho_t - s_t|_{D_t}^2 &= |\nabla \log \rho_t - s_t^* + s_t^* - s_t|_{D_t}^2, \\
&= |\nabla \log \rho_t - s_t^*|_{D_t}^2 + |s_t^* - s_t|_{D_t}^2 + 2(\nabla \log \rho_t - s_t^*) \cdot D_t(s_t^* - s_t), \quad \text{(B.1)} \\
&\geq 2(\nabla \log \rho_t - s_t^*) \cdot D_t(s_t^* - s_t).
\end{aligned}
$$

Hence, we deduce that

$$
\frac{d}{dt} D_{\mathsf{KL}}(\rho_t \mid \rho_t^*) \leq \frac{1}{2} \int_\Omega |s_t(x) - \nabla \log \rho_t(x)|_{D_t(x)}^2 \rho_0(x)dx. \tag{B.2}
$$

$\square$

### B.2 SBTM in the Eulerian frame

The Eulerian equivalent of Proposition 2 can be stated as:

**Proposition B.1** (SBTM in the Eulerian frame)**.** *Assume that the conditions listed in Sec. 1.2 hold. Fix $T \in (0, \infty]$ and consider the optimization problem*

$$
\min_{\{s_t : t \in [0,T)\}} \int_0^T \int_\Omega |s_t(x) - \nabla \log \rho_t(x)|_{D_t(x)}^2 \rho_t(x)dxdt \tag{SBTM2}
$$

*subject to:* $\quad \partial_t \rho_t(x) = -\nabla \cdot (v_t(x)\rho_t(x)), \quad x \in \Omega$

*with $v_t(x) = b_t(x) - D_t(x)s_t(x)$. Then every minimizer of (SBTM2) satisfies $D_t(x)s_t^*(x) = D_t(x)\nabla \log \rho_t^*(x)$ where $\rho_t^* : \Omega \to \mathbb{R}_{>0}$ solves (FPE).*

In words, this proposition states that solving the constrained optimization problem (SBTM2) is equivalent to solving the Fokker-Planck equation (FPE).

*Proof.* The constrained minimization problem (SBTM2) can be handled by considering the extended objective

$$
\int_0^T \int_\Omega \left( |s_t(x) - \nabla \log \rho_t(x)|_{D_t(x)}^2 \rho_t(x) + \mu_t(x) \left( \partial_t \rho_t(x) + \nabla \cdot (v_t(x)\rho_t(x)) \right) \right) dxdt \tag{B.3}
$$

where $v_t(x) = b_t(x) - D_t(x)s_t(x)$ and $\mu_t : \mathbb{R}^d \to \mathbb{R}_{\geq 0}$ is a Lagrange multiplier. The Euler-Lagrange equations associated with (B.3) read

$$
\begin{aligned}
\partial_t \rho_t(x) &= -\nabla \cdot (v_t(x)\rho_t(x)) \\
\partial_t \mu_t(x) &= v_t(x)^\mathsf{T} \nabla \mu_t(x) + |s_t(x)|_{D_t(x)}^2 - |\nabla \log \rho_t|_{D_t(x)}^2 \\
&\quad + 2\nabla \cdot [D_t(x)(s_t(x) - \nabla \log \rho_t(x))], \\
0 &= \mu_T(x), \\
0 &= D_t(x)(s_t(x) - \nabla \log \rho_t(x))\rho_t(x) + \tfrac{1}{2}D_t(x)\nabla \mu_t(x)\rho_t(x)
\end{aligned}
\tag{B.4}
$$

Clearly, these equations will be satisfied if $s_t^*(x) = \nabla \log \rho_t^*(x)$ for all $x \in \Omega$, $\mu_t^*(x) = 0$ for all $x$, and $\rho_t^*$ solves (FPE). This solution is also a global minimizer, because it zeroes the value of the objective. Moreover, all global minimizers must satisfy $D_t(x)s_t^*(x) = D_t(x)\nabla \log \rho_t^*(x)$ ($\rho_t-$almost everywhere), as this is the *only* way to zero the objective.

It is also easy to see that there are no other local minimizers. To check this, we can use the fourth equation to write

$$D_t(x)(s_t(x) - \nabla \log \rho_t(x)) = \tfrac{1}{2}D_t(x)\nabla \mu_t(x).$$

Then,

$$|s_t(x)|^2_{D_t(x)} - |\nabla \log \rho_t(x)|^2_{D_t(x)} = \tfrac{1}{2}\left(s_t(x) + \nabla \log \rho_t(x)\right)^\mathsf{T} D_t(x)\nabla \mu_t(x).$$

This reduces the first three equations to

$$
\begin{aligned}
\partial_t \rho_t(x) &= -\nabla \cdot \big(b_t(x)\rho_t(x) - D_t(x)\nabla \rho_t(x) - \tfrac{1}{2}\rho_t D_t(x)\nabla \mu_t(x)\big) \\
\partial_t \mu_t &= \big(b_t(x) - D_t(x)\nabla \log \rho_t(x) - \tfrac{1}{2}D_t(x)\nabla \mu_t(x)\big)^\mathsf{T}\nabla \mu_t(x) \\
&\quad + \nabla \cdot (D_t(x)\nabla \mu_t(x)) + \tfrac{1}{2}\left(s_t(x) + \nabla \log \rho_t(x)\right)^\mathsf{T} D_t(x)\nabla \mu_t(x). \\
\mu_T(x) &= 0.
\end{aligned}
\tag{B.5}
$$

Since the equation for $\mu_t$ is homogeneous in $\mu_t$ and $\mu_T = 0$, we must have $\mu_t = 0$ for all $t \in [0, T)$, and the equation for $\rho_t$ reduces to (FPE). $\qquad\square$

## B.3 SBTM IN THE LAGRANGIAN FRAME

As stated, Proposition B.1 is not practical, because it is phrased in terms of the density $\rho_t$. The following result demonstrates that the transport map identity (6) can be used to re-express Proposition B.1 entirely in terms of known quantities.

**Proposition 2** (Score-based transport modeling). *Assume that the conditions listed in Sec. 1.2 hold. Define $v_t(x) = b_t(x) - D_t(x)s_t(x)$ and consider*

$$
\begin{aligned}
\dot{X}_t(x) &= v_t(X_t(x)), & X_0(x) &= x, \\
\dot{G}_t(x) &= -[\nabla v_t(X_t(x))]^\mathsf{T} G_t(x) - \nabla\nabla \cdot v_t(X_t(x)), & G_0(x) &= \nabla \log \rho_0(x).
\end{aligned}
\tag{10}
$$

*Then $\rho_t = X_t \sharp \rho_0$ solves (TE), the equality $G_t(x) = \nabla \log \rho_t(X_t(x))$ holds, and for any $T \in [0, \infty)$*

$$D_{\mathsf{KL}}(X_T \sharp \rho_0 \mid \rho_T^*) \leq \frac{1}{2}\int_0^T \int_\Omega |s_t(X_t(x)) - G_t(x)|^2_{D_t(X_t(x))}\,\rho_0(x)dxdt.
\tag{11}$$

*Moreover, if $s_t^*$ is a minimizer of the constrained optimization problem*

$$\min_s \int_0^T \int_\Omega |s_t(X_t(x)) - G_t(x)|^2_{D_t(X_t(x))}\,\rho_0(x)dxdt \quad \text{subject to (10)}
\tag{SBTM}$$

*then $D_t(x)s_t^*(x) = D_t(x)\nabla \log \rho_t^*(x)$ where $\rho_t^*$ solves the Fokker-Planck equation (FPE). The map $X_t^*$ associated to any minimizer is a transport map from $\rho_0$ to $\rho_t^*$, i.e.*

$$x \sim \rho_0 \qquad \text{implies that} \qquad X_t^*(x) \sim \rho_t^*, \qquad \forall t \in [0, T].
\tag{12}$$

*Proof.* Let us first show that $G_t(x) = \nabla \log \rho_t(X_t(x))$ satisfies (10) if $\rho_t = X_t \sharp \rho_0$, i.e. if $\rho_t$ satisfies the transport equation (TE). Since (TE) implies that

$$\partial_t \log \rho_t(x) + v_t(x) \cdot \nabla \log \rho_t(x) = -\nabla \cdot v_t(x),
\tag{B.6}$$

taking the gradient gives

$$\partial_t \nabla \log \rho_t(x) + [\nabla v_t(x)]^\mathsf{T}\nabla \log \rho_t(x) + \nabla\nabla \log \rho_t(x) \cdot v_t(x) = -\nabla\nabla \cdot v_t(x).
\tag{B.7}$$

Therefore $G_t(x) = \nabla \log \rho_t(X_t(x))$ solves

$$
\begin{aligned}
\frac{d}{dt}G_t(x) &= \partial_t \nabla \log \rho_t(X_t(x)) + \nabla\nabla \log \rho_t(X_t(x)) \cdot \frac{d}{dt}X_t(x), \\
&= \partial_t \nabla \log \rho_t(X_t(x)) + \nabla\nabla \log \rho_t(X_t(x)) \cdot v_t(x), \\
&= -\nabla\nabla \cdot v_t(X_t(x)) - [\nabla v_t(X_t(x))]^\mathsf{T}\nabla \log \rho_t(X_t(x)),
\end{aligned}
\tag{B.8}
$$

which recovers the equation for $G_t(x)$ in (10). Hence, the objective in (SBTM) can also be written as

$$\int_0^T \int_\Omega |s_t(X_t(x)) - \nabla \log \rho_t(X_t(x))|^2 \rho_0(x) dx dt$$

$$= \int_0^T \int_\Omega |s_t(x) - \nabla \log \rho_t(x)|^2 \rho_t(x) dx dt \tag{B.9}$$

where the second equality follows from (A.5) if $\rho_t(x)$ satisfies (A.1). Hence, (SBTM) is equivalent to (SBTM2). The bound on $D_{\mathsf{KL}}(X_T \sharp \rho_0 \mid \rho_T^*)$ follows from (9). $\qquad\square$

**Adjoint equations.** In terms of a practical implementation, the objective in (SBTM2) can be evaluated by generating samples $\{x_i\}_{i=1}^n$ from $\rho_0$ and solving the equations for $X_t$ and $G_t$ using the initial conditions $X_0(x_i) = x_i$ and $G_0(x_i) = \nabla \log \rho_0(x_i)$. Note that evaluating this second initial condition only requires one to know $\rho_0$ up to a normalization factor. To evaluate the gradient of the objective, we can introduce equations adjoint to those for $X_t$ and $G_t$. They read, respectively

$$\frac{d}{dt}\theta_t(x) + [\nabla v_t(X_t(x))]^{\mathsf{T}}\theta_t(x) = \eta_t(x) \cdot \nabla\nabla v_t(X_t(x))G_t(x)$$
$$+ \eta_t(x) \cdot \nabla\nabla\nabla v_t(X_t(x))G_t(x)$$
$$+ 2\nabla s_t(X_t(x))(s_t(X_t(x)) - G_t(x)),$$
$$\theta_T(x) = 0 \tag{B.10}$$
$$\frac{d}{dt}\eta_t(x) - \nabla v_t(X_t(x))\eta_t(x) = 2(G_t(x) - s_t(X_t(x))),$$
$$\eta_T(x) = 0.$$

In terms of these functions, the gradient of the objective is the gradient with respect to $s_t(x)$ (or the parameters in this function when it is modeled by a neural network) of the extended objective:

$$L[s_t] = \int_0^T \int_\Omega |s_t(X_t(x)) - G_t(x)|^2 \rho_0(x) dx dt$$
$$+ \int_0^T \int_\Omega \theta_t(x) \cdot \left(\dot{X}_t(x) - v_t(X_t(x))\right) \rho_0(x) dx dt$$
$$+ \int_0^T \int_\Omega \eta_t(x) \cdot \left(\dot{G}_t(x) + [\nabla v_t(X_t(x))]^{\mathsf{T}}G_t(x)\right.$$
$$\left. + \nabla\nabla \cdot v_t(X_t(x))\right)\rho_0(x) dx dt, \tag{B.11}$$

where $v_t(x) = b_t(x) - D_t(x)s_t(x)$.

## B.4 SEQUENTIAL SBTM

Let us restate Proposition 3 for convenience:

**Proposition 3** (Sequential SBTM). *In the same setting as Proposition 2, let $X_t(x)$ solve the first equation in (10) with $v_t(x) = b_t(x) - D_t(x)s_t(x)$. Let $s_t$ be obtained via*

$$\min_{s_t} \int_\Omega \left(|s_t(X_t(x))|^2_{D_t(X_t(x))} + 2\nabla \cdot (D_t(X_t(x))s_t(X_t(x)))\right) \rho_0(x) dx. \tag{SSBTM}$$

*Then, each minimizer $s_t^*$ of (SSBTM) satisfies $D_t(x)s_t^*(x) = D_t(x)\nabla \log \rho_t^*(x)$ where $\rho_t^*$ is the solution to (FPE). Moreover, the map $X_t^*$ associated to $s_t^*$ is a transport map from $\rho_0$ to $\rho_t^*$.*

*Proof.* If $X_t \sharp \rho_0 = \rho_t$, then by definition we have the identity

$$\int_\Omega \left(|s_t(X_t(x))|^2_{D_t(X_t(x))} + 2\nabla \cdot (D_t(X_t(x))s_t(X_t(x)))\right) \rho_0(x) dx$$

$$= \int_\Omega \left(|s_t(x)|^2_{D_t(x)} + 2\nabla \cdot (D_t(x)s_t(x))\right) \rho_t(x) dx. \tag{B.12}$$

This means that the optimization problem in (SSBTM) is equivalent to

$$\min_{s_t} \int_\Omega \left( |s_t(x)|^2_{D_t(x)} + 2\nabla \cdot (D_t(x)s_t(x)) \right) \rho_t(x)dx.$$

All minimizers $s_t^*$ of this optimization problem satisfy $D_t(x)s_t^*(x) = D_t(x)\nabla \log \rho_t(x)$. Hence, by (TE),

$$\partial_t \rho_t(x) = -\nabla \cdot (b_t(x)\rho_t(x) - D_t(x)\nabla \rho_t(x)) \tag{B.13}$$

which recovers (FPE), so that $\rho_t(x) = \rho_t^*(x)$ solves (FPE). $\qquad\square$

## B.5   LEARNING FROM THE SDE

In this section, we show that learning from the SDE alone – i.e., avoiding the use of self-consistent samples – does not provide a guarantee on the accuracy of $\rho_t$. We have already seen in (9) that it is sufficient to control $\int_0^T \int_\Omega |s_t(x) - \nabla \log \rho_t(x)|^2_{D_t} \rho_t^*(x)dxdt$ to control $D_{\mathsf{KL}}(\rho_T \mid \rho_T^*)$. The proof of Proposition 1 shows that control on

$$\int_0^T \int_\Omega |s_t(x) - \nabla \log \rho_t^*(x)|^2_{D_t(x)} \rho_t^*(x)dxdt, \tag{B.14}$$

as would be provided by training on samples from the SDE, does not ensure control on $D_{\mathsf{KL}}(\rho_T \mid \rho_T^*)$. The following proposition shows that control on (B.14) does not guarantee control on $D_{\mathsf{KL}}(\rho_T^* \mid \rho_T)$ either. An analogous result appeared in Lu et al. (2022) in the context of SBDM for generative modeling; here, we provide a self-contained treatment to motivate the use of the sequential SBTM procedure discussed in the main text.

**Proposition B.2.** *Let $\rho_t : \Omega \to \mathbb{R}_{>0}$ solve* (TE)*, and let $\rho_t^* : \Omega \to \mathbb{R}_{>0}$ solve* (FPE)*. Then, the following equality holds*

$$
\begin{aligned}
D_{\mathsf{KL}}(\rho_T^* \mid \rho_T) &= \int_0^T \int_\Omega |s_t(x) - \nabla \log \rho_t^*(x)|^2_{D_t(x)} \rho_t^*(x)dxdt \\
&+ \int_0^T \int_\Omega \left( \nabla \log \rho_t(x) - s_t(x) \right)^\mathsf{T} D_t(x) \left( s_t(x) - \nabla \log \rho_t^*(x) \right) \rho_t^*(x)dxdt.
\end{aligned}
\tag{B.15}
$$

Proposition B.2 shows that minimizing the error between $s_t$ and $\nabla \log \rho_t^*$ on samples of $\rho_t^*$ leaves a remainder term, because in general $\nabla \log \rho_t \neq s_t$. The proof shows that we may obtain the simple upper bound

$$
\begin{aligned}
D_{\mathsf{KL}}(\rho_T^* \mid \rho_T) &\leq \frac{1}{2} \int_0^T \int_\Omega |s_t(x) - \nabla \log \rho_t^*(x)|^2_{D_t(x)} \rho_t^*(x)dxdt \\
&+ \frac{1}{2} \int_0^T \int_\Omega |s_t(x) - \nabla \log \rho_t(x)|^2_{D_t(x)} \rho_t^*(x)dxdt.
\end{aligned}
\tag{B.16}
$$

However, controlling the above quantity requires enforcing agreement between $s_t$ and $\nabla \log \rho_t$ in addition to $s_t$ and $\nabla \log \rho_t^*$; this is precisely the idea of SBTM.

*Proof.* By symmetry, we may replace $\rho_t$ by $\rho_t^*$ in the proof of Proposition 1 to find

$$\frac{d}{dt}D_{\mathsf{KL}}(\rho_t^* \mid \rho_t) = \int \left( \nabla \log \rho_t(x) - \nabla \log \rho_t^*(x) \right)^\mathsf{T} D_t(x) \left( s_t(x) - \nabla \log \rho_t^*(x) \right) \rho_t^*(x)dx$$

Adding and subtracting $s_t(x)$ to the first term in the inner product and expanding gives

$$
\begin{aligned}
\frac{d}{dt}D_{\mathsf{KL}}(\rho_t^* \mid \rho_t) &= \int_\Omega |s_t(x) - \nabla \log \rho_t^*(x)|^2 \rho_t^*(x)dx \\
&+ \int_\Omega \left( \nabla \log \rho_t(x) - s_t(x) \right)^\mathsf{T} D_t(x) \left( s_t(x) - \nabla \log \rho_t^*(x) \right) \rho_t^*(x)dx,
\end{aligned}
\tag{B.17}
$$

Integrating from $0$ to $T$ completes the proof. $\qquad\square$

## B.6 DENOISING LOSS

The following standard trick can be used to avoid computing the divergence of $s_t(x)$:

**Lemma B.3.** *Given $\xi = N(0, I)$, we have*

$$\lim_{\alpha\downarrow 0} \alpha^{-1}\mathbb{E}\big(s_t(x + \alpha\xi) \cdot \xi\big) = \nabla \cdot s_t(x),$$

$$\lim_{\alpha\downarrow 0} \alpha^{-1}\mathbb{E}\big(s_t(x + \alpha\sigma_t(x)\xi) \cdot \sigma_t(x)\xi\big) = \text{tr}\,(D_t(x)\nabla s_t(x)) \tag{B.18}$$

*Proof.* We have

$$\alpha^{-1}s_t(x + \alpha\xi) \cdot \xi = \alpha^{-1}s_t(x) \cdot \xi + (\nabla s_t(x)\xi) \cdot \xi + o(\alpha) \tag{B.19}$$

The expectation of the first term on the right-hand side of this equation is zero; the expectation of the second gives the result in (B.18). Hence, taking the expectation of (B.19) and evaluating the result in the limit as $\alpha \downarrow 0$ gives the first equation in (B.18). The second equation in (B.18) can be proven similarly using $\sigma_t(x)\sigma_t(x)^\mathsf{T} = D_t(x)$. □

Replacing $\nabla \cdot s_t(x)$ in (SSBTM) with the first expression in (B.19) for a fixed $\alpha > 0$ gives the loss

$$\mathcal{L}[s_t] = \mathbb{E}_\xi \left[ \int_\Omega \left( |s_t(X_t(x))|^2 + \frac{2}{\alpha}s_t(X_t(x) + \alpha\xi) \cdot \xi \right) \rho_0(x)dx \right]. \tag{B.20}$$

Evaluating the square term at a perturbed data point recovers the denoising loss of Vincent (2011)

$$\mathcal{L}[s_t] = \mathbb{E}_\xi \left[ \int_\Omega \left| s_t(X_t(x) + \alpha\xi) + \frac{\xi}{\alpha} \right|^2 \rho_0(x)dx \right]. \tag{B.21}$$

We can improve the accuracy of the approximation with a "doubling trick" that applies two draws of the noise of opposite sign to reduce the variance. This amounts to replacing the expectations in (B.18) with

$$\tfrac{1}{2}\alpha^{-1}\mathbb{E}\big[s_t(x + \alpha\xi) \cdot \xi - s_t(x - \alpha\xi) \cdot \xi\big],$$

$$\tfrac{1}{2}\alpha^{-1}\mathbb{E}\big[s_t(x + \alpha\sigma_t(x)\xi) \cdot \sigma_t(x)\xi - s_t(x - \alpha\sigma_t(x)\xi) \cdot \sigma_t(x)\xi\big], \tag{B.22}$$

whose limits as $\alpha \to 0$ are $\nabla \cdot s_t(x)$ and $\text{tr}\,(D_t(x)\nabla s_t(x))$, respectively. In practice, we observe that this approach always helps stabilize training. Moreover, we observe that use of the denoising loss also stabilizes training, so that it is preferable to full computation of $\nabla \cdot s_t(x)$ even when the latter is feasible.

## C GAUSSIAN CASE

Here, we consider the case of an Ornstein-Uhlenbeck (OU) process where the score can be written analytically, thereby providing a benchmark for our approach. The example treated in Section 4.1 with details in Appendix D.1 is a special case of such an OU process with additional symmetry arising from permutations of the particles.

The SDE reads

$$dX_t = -\Gamma_t(X_t - b_t)dt + \sqrt{2}\sigma_t dW_t \tag{C.1}$$

where $X_t \in \mathbb{R}^d$, $\Gamma_t \in \mathbb{R}^{d\times d}$ is a time-dependent positive-definite tensor (not necessarily symmetric), $b_t \in \mathbb{R}^d$ is a time-dependent vector, and $\sigma_t \in \mathbb{R}^{d\times d}$ is a time-dependent tensor. The Fokker-Planck equation associated with (C.1) is

$$\partial_t \rho_t^*(x) = -\nabla \cdot ((\Gamma_t x - b_t)\rho_t^*(x) - D_t\nabla\rho_t^*(x)) \tag{C.2}$$

where $D_t = \sigma_t\sigma_t^\mathsf{T}$. Assuming that the initial condition is Gaussian, $\rho_0 = \mathsf{N}(m_0, C_0)$ with $C_0 = C_0^\mathsf{T} \in \mathbb{R}^{d\times d}$ positive-definite, the solution is Gaussian at all times $t \geq 0$, $\rho_t^* = \mathsf{N}(m_t, C_t)$ with $m_t$ and $C_t = C_t^\mathsf{T}$ solutions to

$$\dot{m}_t = -\Gamma_t(m_t - b_t)$$

$$\dot{C}_t = -\Gamma_t C_t - C_t\Gamma_t^\mathsf{T} + 2D_t \tag{C.3}$$

This implies in particular that

$$\nabla \log \rho_t^*(x) = -C_t^{-1}(x - m_t). \tag{C.4}$$

so that the probability flow equation for $X_t$ and the equation for $G_t$ written in (SBTM2) read

$$\begin{aligned}
\dot{X}_t(x) &= (D_t C_t^{-1} - \Gamma_t)X_t(x) + \Gamma_t b_t - D_t C_t^{-1} m_t, \\
\dot{G}_t(x) &= (\Gamma_t^\mathsf{T} - C_t^{-1} D_t)G_t(x),
\end{aligned} \tag{C.5}$$

with initial condition $X_0(x) = x$ and $G_0(x) = \nabla \log \rho_0(x) = -C_0^{-1}(x - m_0)$. It is easy to see that with $x \sim \rho_0 = \mathsf{N}(m_0, C_0)$ we have $X_t(x) \sim \rho_t^* = \mathsf{N}(m_t, C_t)$ since, from the first equation in (C.5), the mean and variance of $X_t$ satisfy (C.3). Similarly, when $x \sim \rho_0 = \mathsf{N}(m_0, C_0)$, $G_0(x) \sim N(0, C_0^{-1})$, so that $G_t(x) \sim \mathsf{N}(0, C_t^{-1})$ because the second equation in (C.5) is linear and hence preserves Gaussianity. Moreover, $\mathbb{E}_0 G_t(x) = 0$ and $B_t = B_t^\mathsf{T} = \mathbb{E}_0[G_t(x)G_t^\mathsf{T}(x)]$ satisfies

$$\frac{d}{dt}B_t = (\Gamma_t^\mathsf{T} - C_t^{-1}D_t)B_t + B_t(\Gamma_t - D_t C_t^{-1}) \tag{C.6}$$

The solution to this equation is $B_t = C_t^{-1}$ since substituting this ansatz into (C.6) gives the equation for $C_t^{-1}$ that we can deduce from (C.3)

$$\frac{d}{dt}C_t^{-1} = C_t^{-1}\dot{C}_t C_t^{-1} = -C_t^{-1}\Gamma_t - \Gamma_t^\mathsf{T} C_t^{-1} + 2C_t^{-1}D_t C_t^{-1}. \tag{C.7}$$

Note that if $\Gamma_t = \Gamma$, $b_t = b$, and $D_t = D$ are all time-independent, then $\lim_{t\to\infty} \rho_t = N(m_\infty, C_\infty)$ with $m_\infty = b$ and $C_\infty$ the solution to the Lyapunov matrix equation

$$\Gamma C_\infty + C_\infty \Gamma^\mathsf{T} = 2D. \tag{C.8}$$

This means that at long times the coefficients at the right-hand sides of (C.5) also settle on constant values. However, $X_t$ and $G_t$ do not necessarily stop evolving; one situation where they too tend to fix values is when the OU process is in detailed balance, i.e. when $\Gamma = DA$ for some $A = A^\mathsf{T} \in \mathbb{R}^{d \times d}$ positive-definite. In that case, the solution to (C.8) is $C_\infty = A^{-1}$ and it is easy to see that at long times the right hand sides of (C.5) tend to zero.

*Remark* C.1. This last conclusion is actually more generic than for a simple OU process. For any SDE in detailed balance, i.e. that can be written as

$$dX_t = -D(X_t)\nabla U(X_t)dt + \nabla \cdot D(X_t)dt + \sqrt{2}\sigma_t(X_t)dW_t \tag{C.9}$$

where $U : \mathbb{R}^d \to \mathbb{R}_{>0}$ is a $C^2$-potential such that $Z = \int_{\mathbb{R}^d} e^{-U(x)}dx < \infty$, we have that $\lim_{t\to\infty} \rho_t(x) = Z^{-1}e^{-U(x)}$, and the corresponding flows $X_t$ and $G_t$ eventually stop as $t \to \infty$. In this case, $\rho_t$ follows gradient descent in $W_2$ over the energy

$$E[\rho] = \int_{\mathbb{R}^d} (U(x) + \log \rho(x))\rho(x)dx \tag{C.10}$$

The unique PDF minimizing this energy is $Z^{-1}e^{-U(x)}$, and as $t \to \infty$ $X_t$ converges towards a transport map between the initial $\rho_0$ and $Z^{-1}e^{-U(x)}$.

# D    Experimental details and additional examples

All numerical experiments were performed in `jax` using the `dm-haiku` package to implement the networks and the `optax` package for optimization.

## D.1    Harmonically interacting particles in a harmonic trap

**Network architecture** Both the single-particle energy $U_{\theta_t,1} : \mathbb{R}^d \to \mathbb{R}$ and two-particle interaction energy $U_{\theta_t,2} : \mathbb{R}^d \times \mathbb{R}^d \to \mathbb{R}$ are parameterized as single hidden-layer neural networks with the `swish` activation function (Ramachandran et al., 2017) and `n_hidden = 100` hidden neurons. The hidden layer biases are initialized to zero while the hidden layer weights are initialized from

a truncated normal distribution with variance $1/\texttt{fan\_in}$, following the guidelines recommended in (Ioffe & Szegedy, 2015).

**Optimization** The Adam (Kingma & Ba, 2017) optimizer is used with an initial learning rate of $\eta = 10^{-4}$ and otherwise default settings. At time $t = 0$, the analytical relative loss

$$L[s_0] = \frac{\int |s_0(x) - \nabla \log \rho_0(x)|^2 \rho_0(x) dx}{\int |\nabla \log \rho_0(x)|^2 \rho_0(x) dx} \tag{D.1}$$

is minimized to a value less than $10^{-4}$ using knowledge of the initial condition $\rho_0 = \mathsf{N}\left(\beta_0, \sigma_0^2 I\right)$ with $\sigma_0 = 0.25$. In (D.1), the expectation with respect to $\rho_0$ is approximated by an initial set of samples $x_j = \left(x_j^{(1)}, x_j^{(2)}, \ldots, x_j^{(N)}\right)^\mathsf{T}$ with $j = 1, \ldots, n$ drawn from $\rho_0$. In the experiments, we set $n = 100$. We set the physical timestep $\Delta t = 10^{-3}$ and take $\texttt{n\_opt\_steps} = 25$ steps of Adam until the norm of the gradient is below $\texttt{gtol} = 0.1$.

**Analytical moments** First define the mean, second moment, and covariance according to

$$m_t^{(i)} = \mathbb{E}\big[X_t^{(i)}\big],$$
$$M_t^{(ij)} = \mathbb{E}\big[X_t^{(i)}\big(X_t^{(j)}\big)^\mathsf{T}\big],$$
$$C_t^{(ij)} = M^{(ij)} - m^{(i)}\big(m^{(j)}\big)^\mathsf{T}.$$

It is straightforward to show that the mean and covariance obey the dynamics

$$\dot{m}_t^{(i)} = -(m_t^{(i)} - \beta_t) + \frac{\alpha}{N} \sum_{k=1}^{N} \left(m_t^{(i)} - m_t^{(k)}\right), \tag{D.2}$$

$$\dot{C}_t^{(ij)} = -2(1-\alpha)C_t^{(ij)} + 2DI\delta_{ij} - \frac{\alpha}{N}\sum_{k=1}^{N}\left(C_t^{(kj)} + C_t^{(ik)}\right) \tag{D.3}$$

Because the particles are indistinguishable so long as they are initialized from a distribution that is symmetric with respect to permutations of their labeling, the moments will satisfy the ansatz

$$m_t^{(i)} = \bar{m}(t), \quad i = 1, \ldots, N \tag{D.4}$$

$$C_t^{(ij)} = C_d(t)\delta_{ij} + C_o(t)(1 - \delta_{ij}), \quad i, j = 1, \ldots, N. \tag{D.5}$$

The dynamics for the vector $\bar{m} : \mathbb{R}_{\geq 0} \to \mathbb{R}^{\bar{d}}$, as well as the matrices $C_d : \mathbb{R}_{\geq 0} \to \mathbb{R}^{\bar{d} \times \bar{d}}$ and $C_o : \mathbb{R}_{\geq 0} \to \mathbb{R}^{\bar{d} \times \bar{d}}$ can then be obtained from (D.2) and (D.3) as

$$\dot{\bar{m}} = \beta_t - \bar{m},$$
$$\dot{C}_d = 2(\alpha - 1)C_d - 2\frac{\alpha}{N}\left(C_d + (n-1)C_o\right) + 2DI,$$
$$\dot{C}_o = 2(\alpha - 1)C_o - 2\frac{\alpha}{N}\left(C_d + (n-1)C_o\right).$$

For a given $\beta : \mathbb{R} \to \mathbb{R}^{\bar{d}}$, these equations can be solved analytically in Mathematica as a function of time, giving the mean $m_t = \bar{m}(t) \otimes 1_N \in \mathbb{R}^{N\bar{d}}$ and covariance $C_t = (C_d(t) - C_o(t)) \otimes I_{N \times N} + C_o(t) \otimes \left(1_N 1_N^\mathsf{T}\right) \in \mathbb{R}^{N\bar{d} \times N\bar{d}}$. Because the solution is Gaussian for all $t$, we then obtain the analytical solution to the Fokker-Planck equation $\rho_t^* = \mathsf{N}(m_t, C_t)$ and the corresponding analytical score $-\nabla \log \rho_t^*(x) = C_t^{-1}(x - m_t)$.

**Potential structure** Here, we show that the potential for this example lies in the class of potentials described by (15). From Equation D.5, we have a characterization of the structure of the covariance matrix $C_t$ for the analytical potential $U_t(x) = \frac{1}{2}(x - m_t)^\mathsf{T} C_t^{-1}(x - m_t)$. In particular, $C_t$ is block circulant, and hence is block diagonalized by the roots of unity (the block discrete Fourier transform). That is, we may take a "block eigenvector" of the form $\omega_k = \left(I_{\bar{d} \times \bar{d}}\rho^k, I_{\bar{d} \times \bar{d}}\rho^{2k}, \ldots, I_{\bar{d} \times \bar{d}}\rho^{(N-1)k}\right)^\mathsf{T}$ with $\rho = \exp(-2\pi i/N)$ for $k = 0, \ldots N - 1$. By direct calculation, this block diagonalization leads to two distinct block eigenmatrices,

$$C_t = V \begin{pmatrix} C_d(t) + (N-1)C_o(t) & 0 & 0 & \ldots & 0 \\ 0 & C_d(t) - C_o(t) & 0 & \ldots & 0 \\ 0 & 0 & \ddots & \ldots & 0 \\ 0 & 0 & 0 & \ldots & C_d(t) - C_o(t) \end{pmatrix} V^{-1}$$

where $V \in \mathbb{R}^{N\bar{d} \times N\bar{d}}$ denotes the matrix with block columns $\omega_k$. The inverse matrix $C_t^{-1}$ then must similarly have only two distinct block eigenmatrices given by $(C_d(t) + (N-1)C_o(t))^{-1}$ and $(C_d(t) - C_o(t))^{-1}$. By inversion of the block Fourier transform, we then find that

$$\left(C_t^{-1}\right)^{(ij)} = \bar{C}_d \delta_{ij} + \bar{C}_o(1 - \delta_{ij})$$

for some matrices $\bar{C}_d, \bar{C}_o$. Hence, by direct calculation

$$
\begin{aligned}
(x - m_t)^\mathsf{T} C_t^{-1} (x - m_t) &= \sum_{i,j}^N \left(x^{(i)} - m_t^{(i)}\right)^\mathsf{T} \left(C_t^{-1}\right)^{(ij)} \left(x^{(j)} - m_t^{(j)}\right) \\
&= \sum_{i,j}^N \left(x^{(i)} - \bar{m}(t)\right)^\mathsf{T} \left(\bar{C}_d \delta_{ij} + \bar{C}_o(1 - \delta_{ij})\right) \left(x^{(j)} - \bar{m}(t)\right) \\
&= \sum_i^N \left(x^{(i)} - \bar{m}(t)\right)^\mathsf{T} \bar{C}_d \left(x^{(i)} - \bar{m}(t)\right)^\mathsf{T} \\
&\qquad + \sum_{i \neq j}^N \left(x^{(i)} - \bar{m}(t)\right)^\mathsf{T} \bar{C}_o \left(x^{(j)} - \bar{m}(t)\right)
\end{aligned}
$$

Above, we may identify the first term in the last line as $\sum_{i=1}^N U_1(x^{(i)})$ and the second term in the last line as $\frac{1}{N} \sum_{i \neq j}^N U_2(x^{(i)}, x^{(j)})$. Moreover, $U_2(\cdot, \cdot)$ is symmetric with respect to its arguments.

**Analytical Entropy** For this example, the entropy can be computed analytically and compared directly to the learned numerical estimate. By definition,

$$
\begin{aligned}
s_t &= -\int_{\mathbb{R}^{N\bar{d}}} \log \rho_t(x) \rho_t(x) dx, \\
&= -\int_{\mathbb{R}^{N\bar{d}}} \left(-\frac{N\bar{d}}{2} \log(2\pi) - \frac{1}{2} \log \det C_t - \frac{1}{2}(x - m_t)^\mathsf{T} C_t^{-1}(x - m_t)\right) \rho_t(x) dx, \\
&= \frac{N\bar{d}}{2} \left(\log(2\pi) + 1\right) + \frac{1}{2} \log \det C_t.
\end{aligned}
$$

**Additional figures** Images of the learned velocity field and potential in comparison to the corresponding analytical solutions can be found in Figures D.1 and D.2, respectively. Further detail can be found in the corresponding captions. We stress that the two-dimensional images represent single-particle slices of the high-dimensional functions.

## D.2 Soft spheres in an anharmonic trap

**Network architecture** Both potential terms $U_{\theta_t,1}$ and $U_{\theta_t,2}$ are modeled as four hidden-layer deep fully connected networks with `n_hidden` $= 32$ neurons in each layer. The initialization is identical to Appendix D.2.

**Optimization and initialization** The Adam optimizer is used with an initial learning rate of $\eta = 5 \times 10^{-3}$ and otherwise default settings. At time $t = 0$, the loss (D.1) is minimized to a value less than $10^{-4}$ over $n$ samples $x_{0,j} \sim \mathsf{N}(\beta_0, \sigma_0^2 I)$ with $\sigma_0 = 0.5$ and $n = 1000$, similar to Appendix D.2. After this initial optimization, 100 steps of the SDE (17) are taken in artificial time $\tau$ with fixed physical $t = 0$ to ensure that no spheres are overlapping at initialization. Past this initial stage, the denoising loss is used with a noise scale $\sigma = 0.025$. The loss is minimized by taking `n_opt_steps` $= 25$ steps of Adam until the norm of the gradient is below `gtol` $= 0.5$. The physical timestep is set to $\Delta t = 10^{-3}$.

**Additional figures** A depiction of the one-particle potential, estimated as the negative logarithm of the one-particle PDF obtained via kernel density estimation, can be found in Figure D.3 (for further details, see the caption).

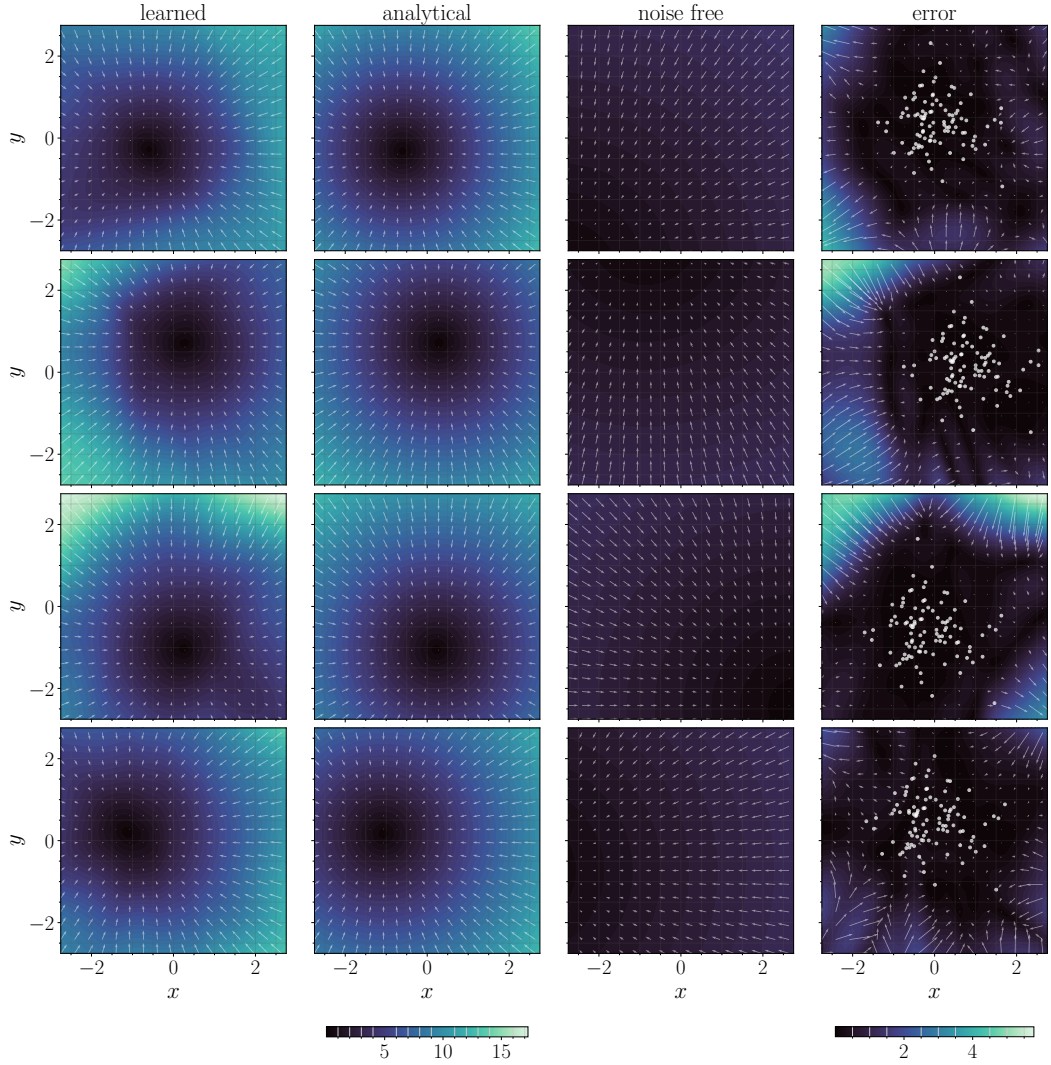

Figure D.1: *A system of $N = 50$ harmonically interacting particles in a harmonic trap: slices of the high-dimensional velocity field.* Cross sections of the velocity field for $N = 50$ harmonically interacting particles in a moving harmonic trap. Columns depict the learned, analytical, noise-free, and error between the learned and analytical velocity fields, respectively. Rows indicate different time points, corresponding to $t = 1.25, 2.5, 3.75$, and $5.0$, respectively. Each velocity field is plotted as a function of a single particle's coordinate (denoted as $x$ and $y$); all other particle coordinates are fixed to be at the location of a sample. Color depicts the magnitude of the velocity field while arrows indicate the direction. Learned, analytical, and noise-free share a colorbar for direct comparison; the error occurs on a different scale and is plotted with its own colorbar. White circles in the error plot indicate samples projected onto the $xy$ plane; locations of low error correlate well with the presence of samples.

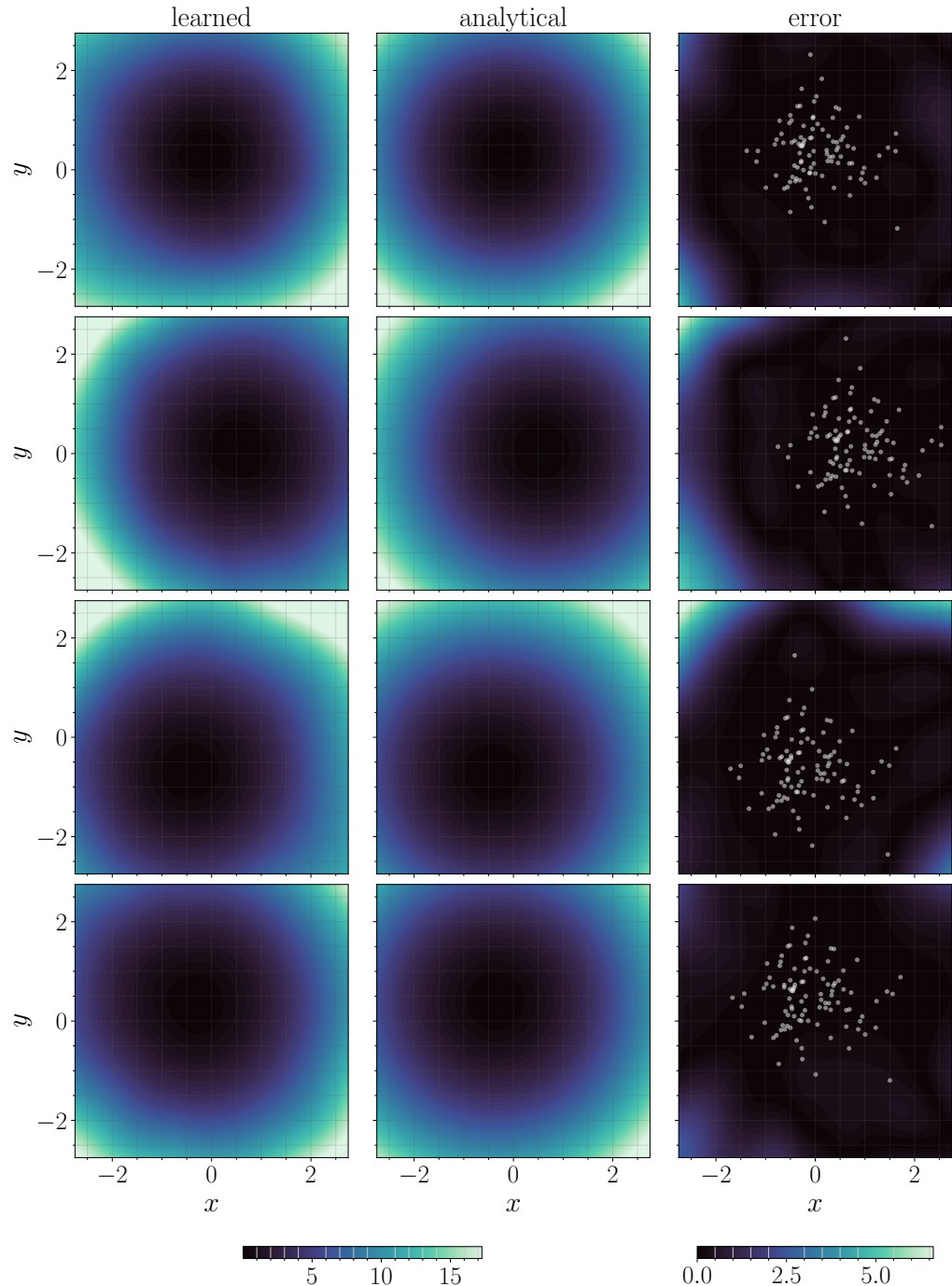

Figure D.2: *A system of $N = 50$ harmonically interacting particles in a harmonic trap: slices of the high-dimensional potential.* Cross sections of the potential field $U_{\theta_t}(x)$ computed via (15). Columns depict the learned, analytical, and error between the learned and analytical, respectively. Rows indicate distinct time points, corresponding to $t = 1.25, 2.5, 3.75$, and $5.0$, respectively. As in Figure D.1, each potential field is plotted as a function of a single particle's coordinate (denoted as $x$ and $y$) with other particle coordinates fixed on a sample. All potentials are normalized via an overall shift so that the minimum value is zero. White circles in the error plot indicate samples from the learned system projected onto the $xy$ plane.

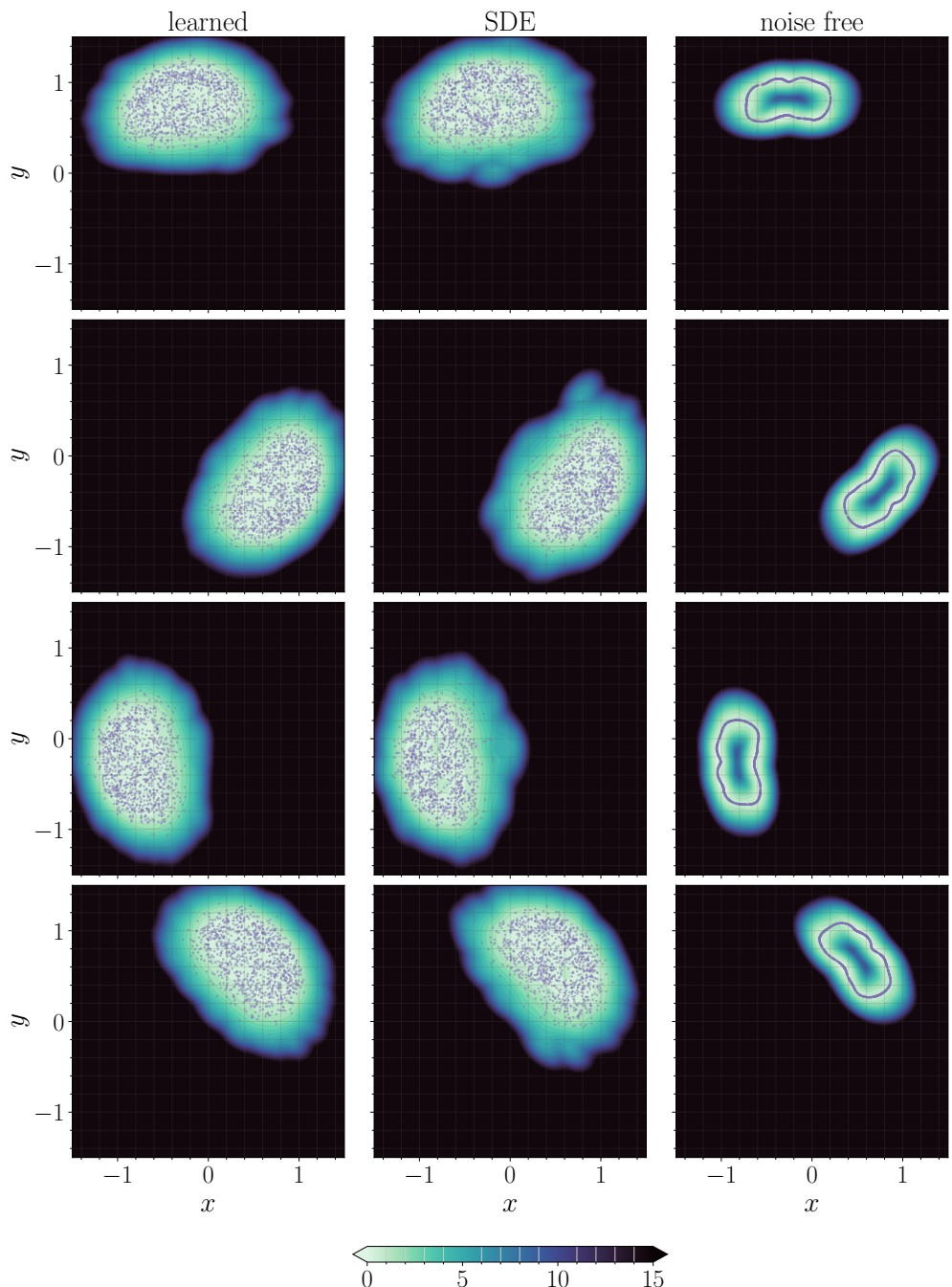

Figure D.3: *A system of $N = 5$ soft-sphere particles in an anharmonic trap: one-particle potential.* Cross sections of the one-particle potential field $U(x) = -\log \rho_{\text{KDE}}(x)$ where $\rho_{\text{KDE}}$ denotes a kernel density estimate of the one-particle density obtained by pooling all particles and treating them as equivalent two-dimensional samples, shown relative to the moving mean. Columns depict the learned, SDE, and noise free systems, respectively. Purple dots indicate samples from the corresponding system. Rows indicate distinct time points, corresponding to $t = 1.25, 2.5, 3.75$, and $4.95$, respectively. All potentials are normalized via an overall shift so that the minimum value is zero, and are clipped to a maximum value of $15$. The learned and SDE potentials match well, while the noise free KDE becomes too peaked and develops a spurious maximum that causes the particles to align in a ring.

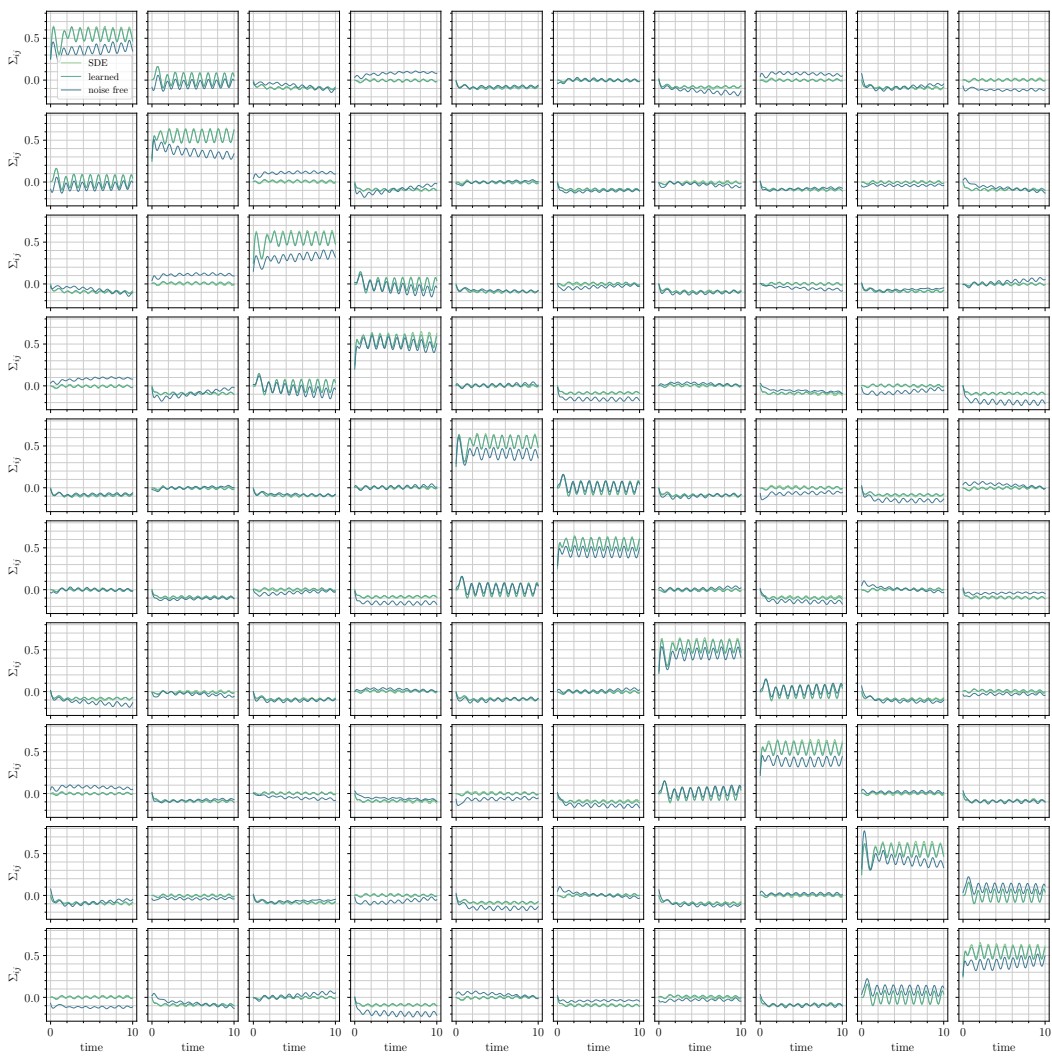

Figure D.4: *A system of $N = 5$ soft-sphere particles in an anharmonic trap: moments.* All components of the covariance matrix over time for the circular trap motion.

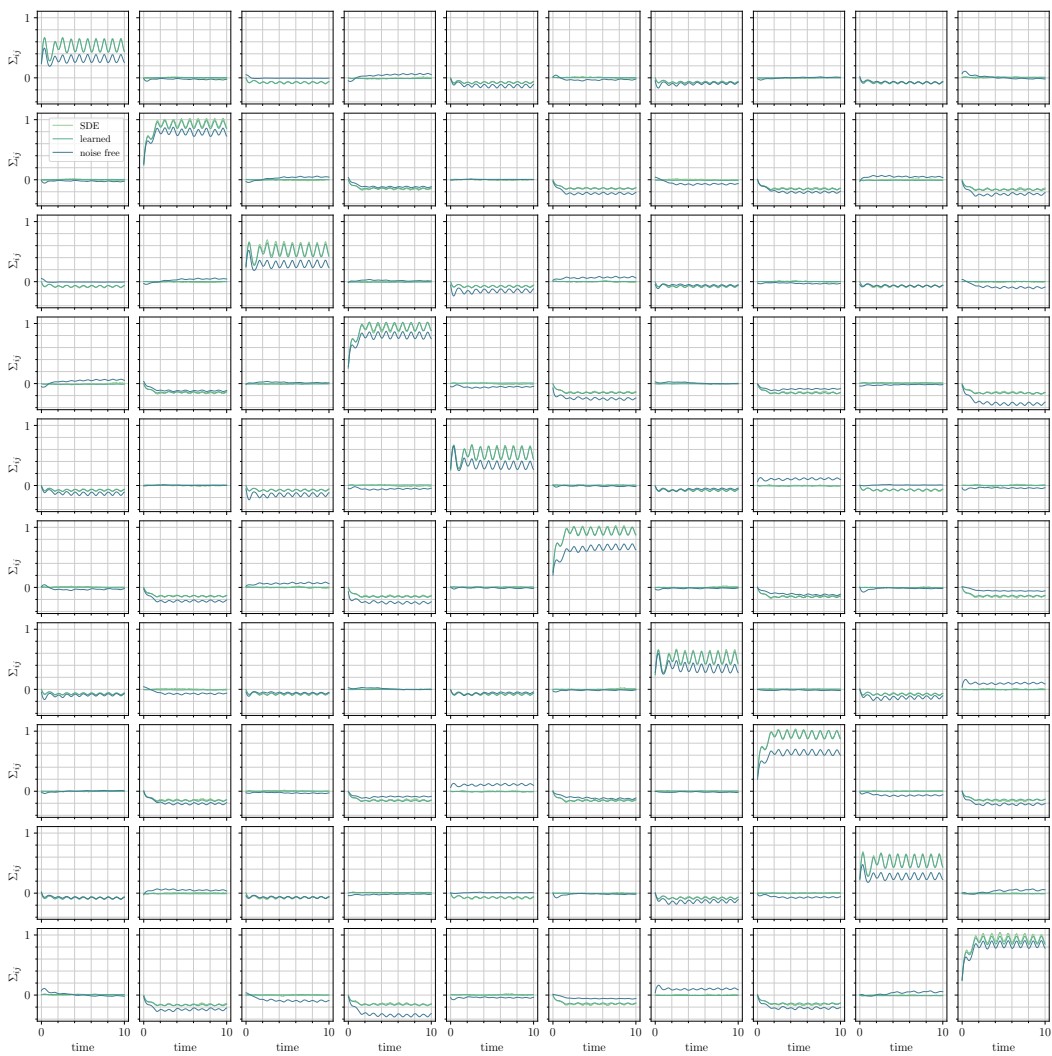

Figure D.5: *A system of* $N = 5$ *soft-sphere particles in an anharmonic trap: moments.* All components of the covariance matrix over time for the linear trap motion.

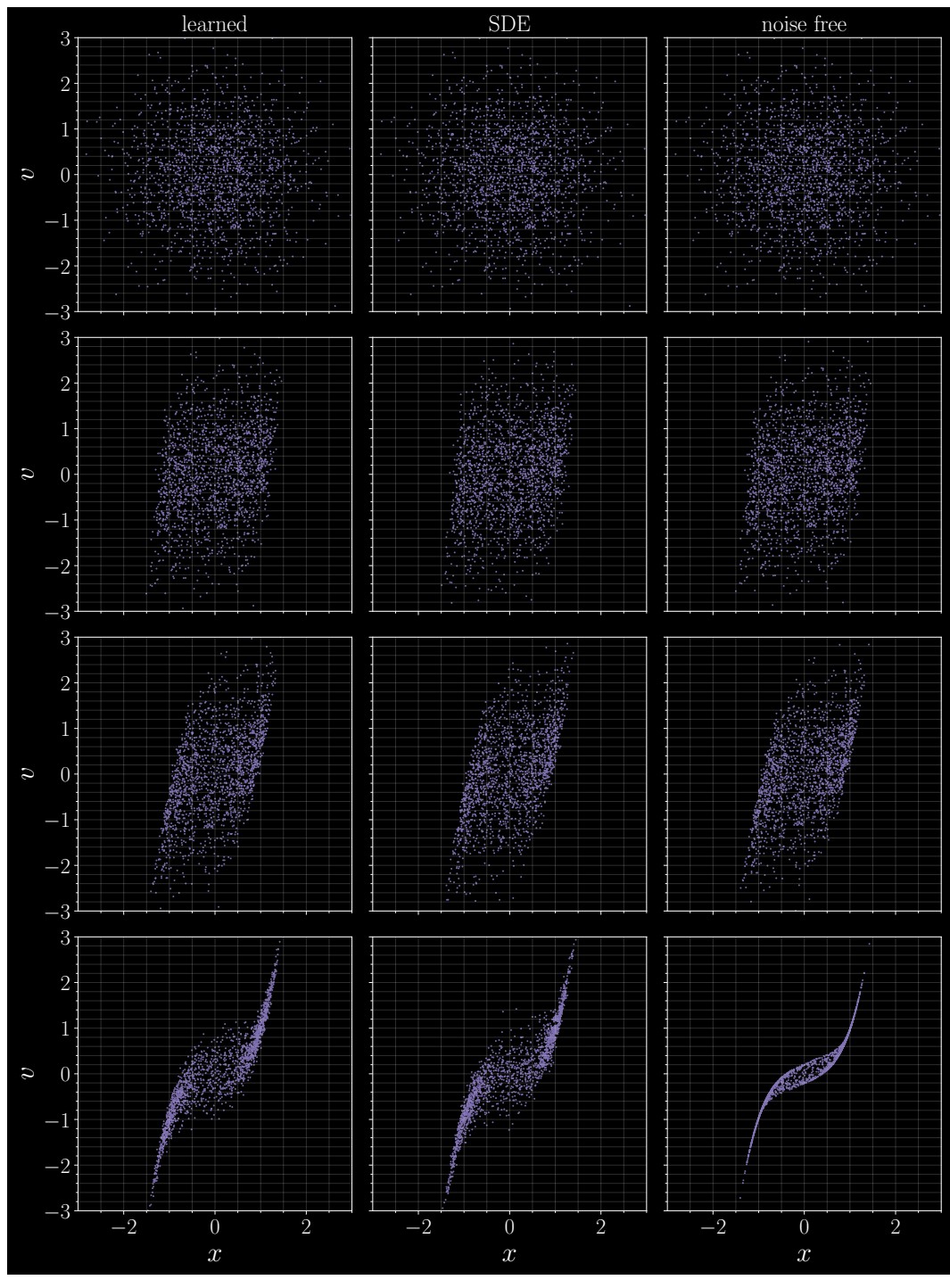

Figure D.6: *An active swimmer: sample trajectories.* Samples in the $xv$ plane. Columns denote solution type and rows indicate snapshots in time ($t = 0, 0.25, 0.5, 3.0$, respectively). The learned and SDE systems develop bimodality while the noise free system collapses with time and does not correctly capture the variance.

## D.3  AN ACTIVE SWIMMER

Here, we study an "active swimmer" model that describes the motion of a particle in an anharmonic trap with a preference to travel in a noisy direction. The system is two-dimensional, and is given by the stochastic differential equation for the position $x$ and velocity $v$

$$dx = \left(-x^3 + v\right) dt,$$
$$dv = -\gamma v dt + \sqrt{2\gamma D} dW_t. \tag{D.6}$$

Despite its low-dimensionality, (D.6) exhibits convergence to a non-equilibrium statistical steady state in which the probability current $j_t(x) = v_t(x)\rho_t(x)$ is non-zero.

**Setup** We set $\gamma = 0.1$ and $D = 1.0$. Because noise only enters the system through the velocity variable $v$ in (D.6), the score can be taken to be one-dimensional. This is equivalent to learning the score only in the range of the rank-deficient diffusion matrix. We parameterize the score directly $s_t : \mathbb{R}^2 \to \mathbb{R}$ using a three-hidden layer neural network with `n_hidden = 32` neurons per hidden layer.

**Optimization and initialization** The network initialization is identical to the previous two experiments. The physical timestep is set to $\Delta t = 10^{-3}$. The Adam optimizer is used with an initial learning rate of $\eta = 10^{-4}$. At time $t = 0$ the loss (D.1) is minimized to a tolerance of $10^{-4}$ over $n = 5000$ samples drawn from an initial distribution $\mathsf{N}(0, \sigma_0^2 I)$ with $\sigma_0 = 1$. The denoising loss is used with a noise scale $\sigma = 0.05$, using `n_opt_steps = 25` steps of Adam until the norm of the gradient is below `gtol = 0.5`.

**Results** Depictions of the sample trajectories $\{x_i(t), v_i(t)\}_{i=1}^n$ in phase space are shown in Figure D.6. The trajectories demonstrate that the distribution of the learned samples qualitatively matches the distribution of the SDE samples. The noise-free system grows increasingly and overly compressed with time. The learned velocity field effectively captures a non-zero rotational steady-state current that qualitatively matches the current of the SDE but enjoys more interpretable sample trajectories.

A movie of the motion of the samples $(x_i, v_i)$ in phase space can be seen here. The movie highlights convergence of the learned solution to a non-zero steady-state probability current that qualitatively matches that of the SDE. By contrast, the noise-free system becomes increasingly concentrated with time, failing to accurately capture the current. Figure D.7 depicts the learned velocity field $v_t(x) = b_t(x) - Ds_t(x)$. The figure highlights the structure of the steady-state current, which contains an elliptical region with closed orbits. The elliptical region remains roughly fixed in size as time proceeds, while the orbits of the noise-free system in Figure D.8 become increasing compressed. Kernel density estimation demonstrates that an estimated PDF for the samples of learned solution qualitatively matches that of the SDE (Figure D.9).

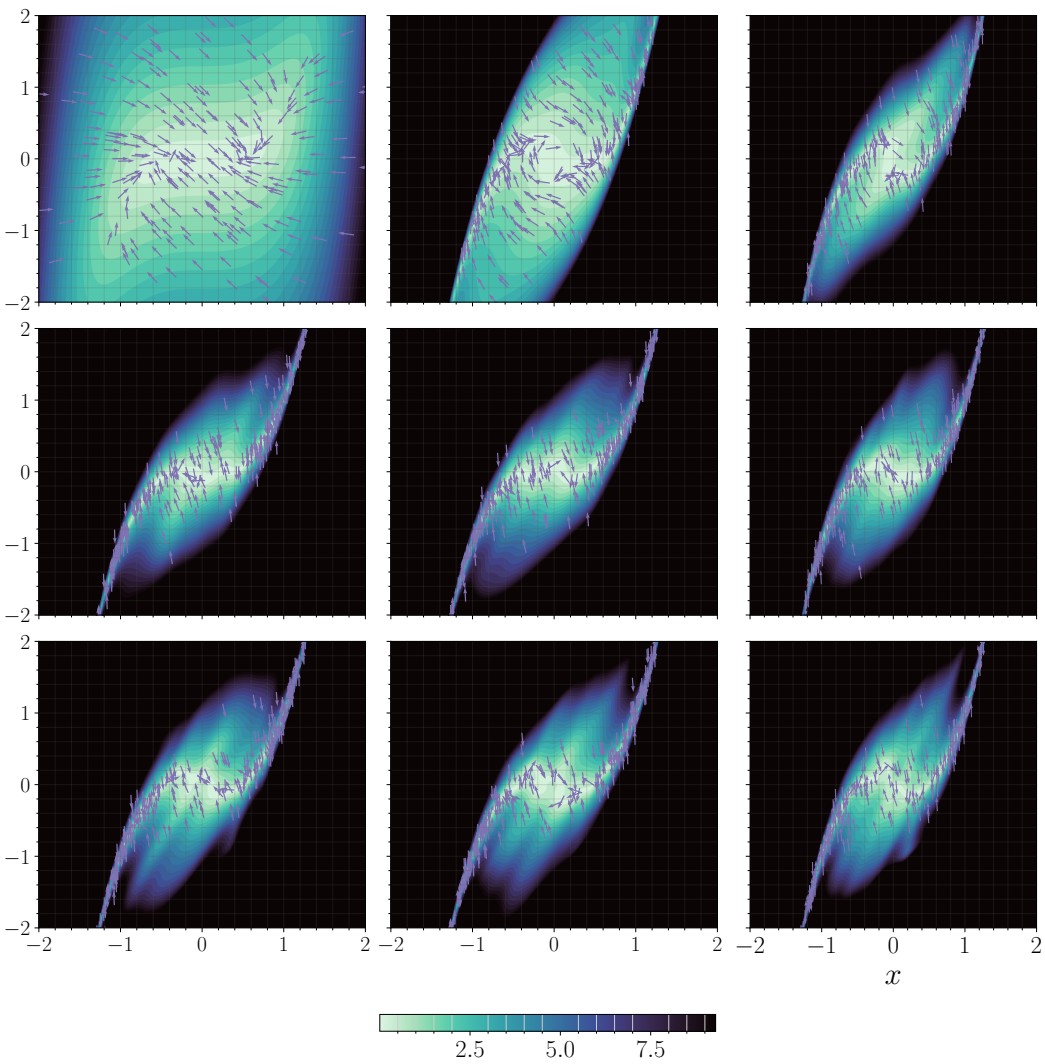

Figure D.7: *An active swimmer: learned velocity.* The learned velocity field (right-hand side of (5)) for the active swimmer example. Color indicates the magnitude of the velocity field computed on a grid, while arrows indicate the direction of the velocity field on samples. Time corresponds to progressing in the grid along columns from the top-left to the bottom-right image ($t = k \times .75$ with $k$ the image number, zero-indexed). The learned velocity field converges to closed streamlines that enforce a nonzero steady-state current.

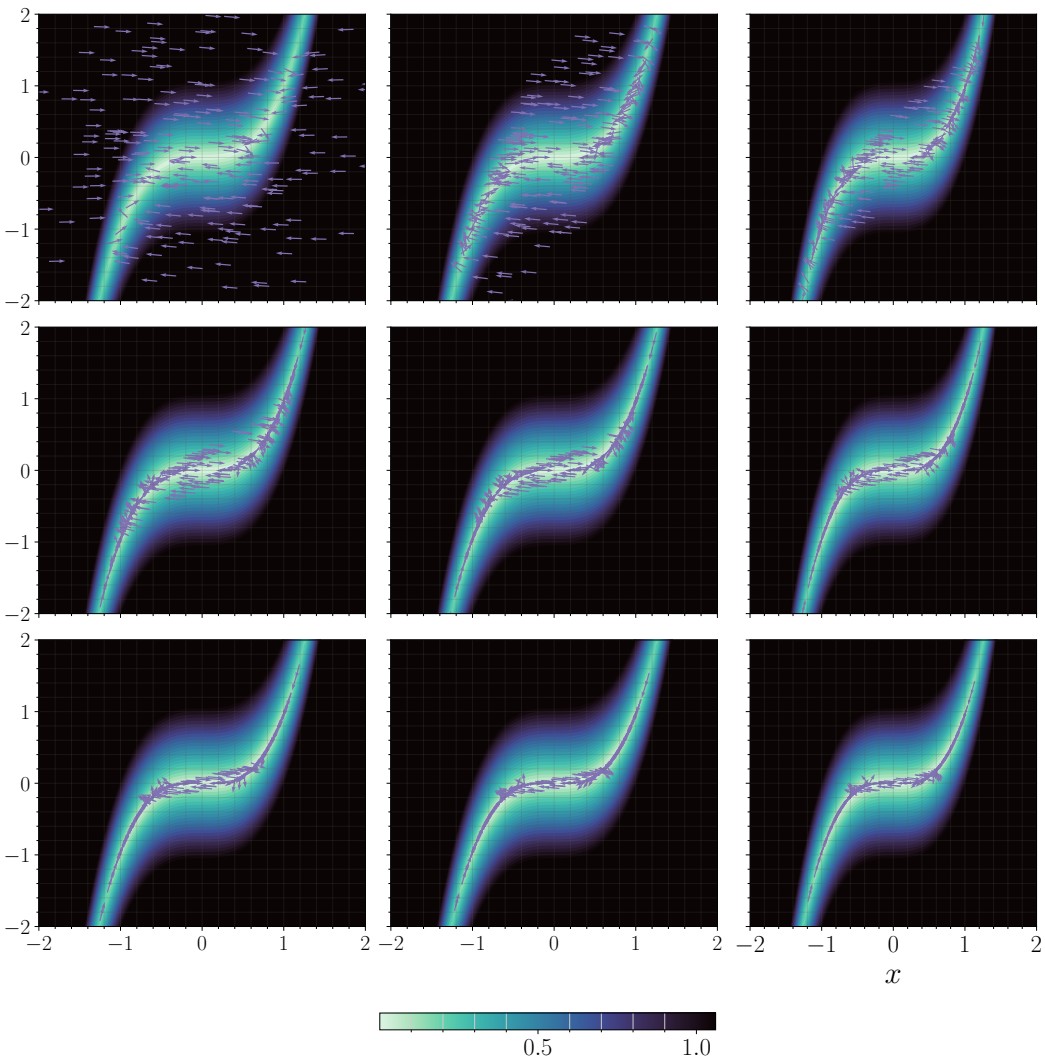

Figure D.8: *An active swimmer: noise free velocity.* Noise free velocity field. As in Figure D.7, color indicates the magnitude of the velocity field while arrows indicate the direction, and time corresponds to progressing in the grid along columns from the top-left to the bottom-right image ($t = k \times .75$ with $k$ the image number, zero-indexed). The velocity field in the noise-free case incorrectly pushes the swimmers to lie along a thin band.

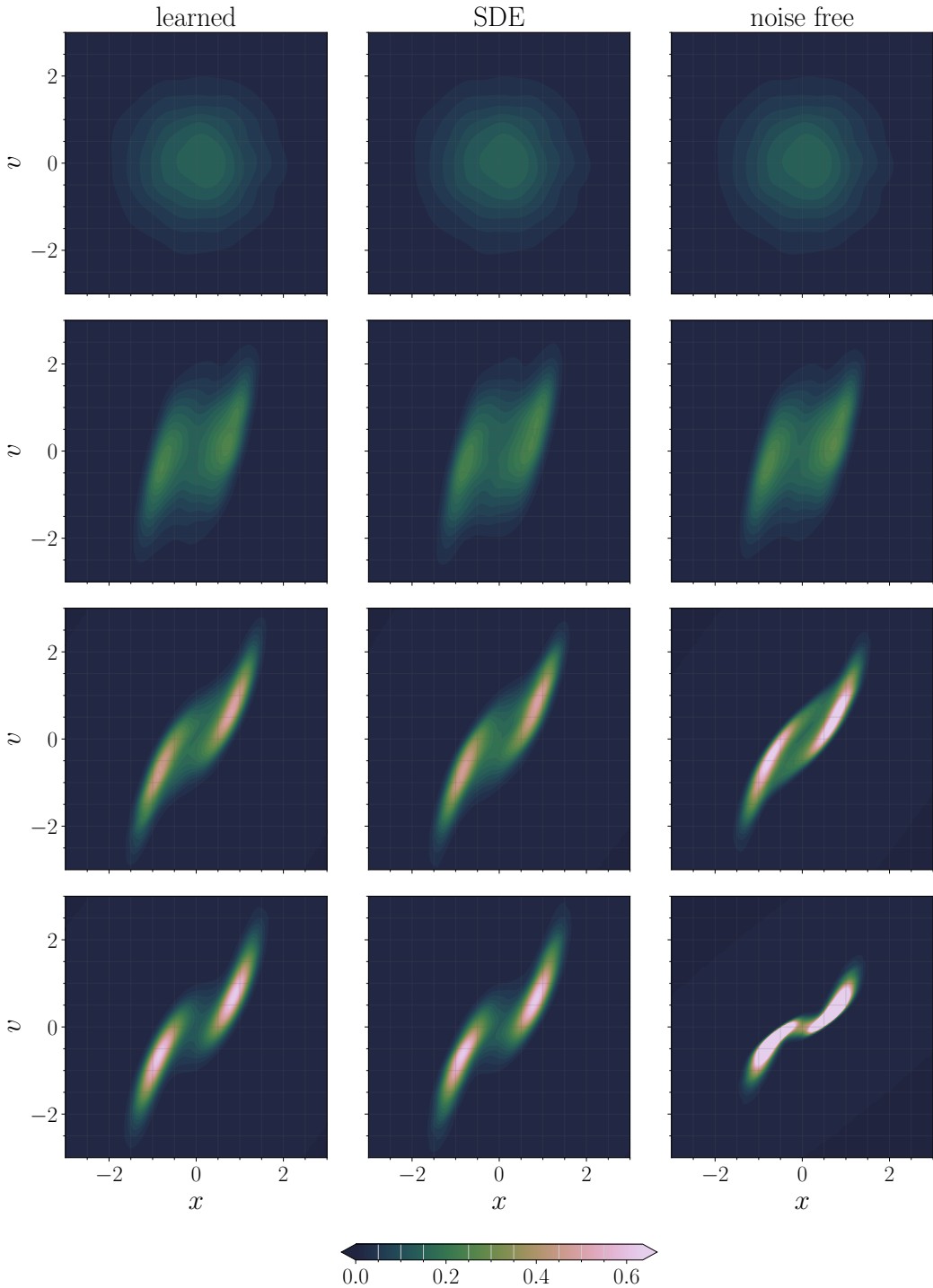

Figure D.9: *An active swimmer: density.* PDFs computed via kernel density estimation in the $xv$ plane. Columns denote solution type and rows denote snapshots in time ($t = 0, 0.5, 1.5, 6.0$, respectively). Similar to the samples presented in Figure D.6, the KDE reveals bimodality in the probability density due to the presence of the particle velocity field. The noise free system becomes too concentrated and does not accurately capture the shape of the SDE and learned solutions, while the SDE and learned solutions are nearly identical.

