# OpenReview forum: "Probability flow solution of the Fokker-Planck equation"
_ICLR.cc/2023/Conference — Submitted to ICLR 2023_

### Official Review · Reviewer_LTVr · 2022-10-21

**Confidence:** 4
**Correctness:** 3
**Technical Novelty And Significance:** 2
**Empirical Novelty And Significance:** 2
**Recommendation:** 5

**Clarity, Quality, Novelty And Reproducibility:**

The clarity of the paper is good. However, the source code is not provided.


**Strength And Weaknesses:**

## Strengths:
* The idea of learning the score from particles while simulating particles along the probability flow of the Fokker-Planck equation using the score is a very interesting one.
* Theoretically there are some interesting perspectives on viewing SBTM in Lagrangian or Eulerian frames, and the KL divergence with respect to the true flow can be controlled by the error of the score.


## Weaknesses:
* The motivation for learning the score during the flow is not super clear to me when you can simply simulate the SDE. The authors suggest you can obtain more quantities like the density of the flow, yet to me, the only new quantity you obtain is the score. And to obtain the score, you could also just simulate the SDE using Euler, and learn the score as a post-processing step using (7), i.e., I don't see any practical benefit of learning the flow while simulating the particles which can be very costly. The only way I can think of to obtain the density is to keep the *entire history* of the particles (which can be very memory intensive) and then compute the density using (5), but this is not demonstrated in any experiment.
* I found Proposition 1 quite misleading and not relevant to the actual algorithm proposed. What is the role of $G_t$? Isn't it just the true score? The constrained optimization in (SBTM) formulation is never actually done (it is mentioned only in passing that it can be solved using the adjoint method). The function $\lambda$ appears to not matter as it is simply taken to be a dirac delta in the sequential SBTM (wouldn't this also break the assumptions in Proposition 1? Dirac deltas are not positive functions).
* Equation (seqSBTM) seems identical to me as (7). It is possible that I entirely misunderstood, but to me Algorithm 1 has very little to do with the theories developed in Section 3 and can be simply obtained by applying (7) to get the score and then integrating the ODE (3)+(4).
* I found the experiments section quite weak. Despite the main selling point of the proposed algorithm being its ability to access quantities like the density, it is never demonstrated in what occasions these quantities are useful. It is only verified that these quantities are accurate. What downstream applications can these quantities facilitate? The authors should also compare with the naive solution which is to simulate the corresponding SDE using Euler scheme and then estimate these quantities as postprocessing steps, such as using kernel density estimation. I would also hope to see more interesting instances of Fokker-Planck equations rather than the two toyish examples shown here.


**Summary Of The Paper:**

This paper proposes score-based transport modeling (SBTM), a particle-based method to solve the Fokker-Planck equation. This is based on the transport map approach (TE) where the key challenge is to estimate the score of the distribution at the current time. Such a challenge is resolved by learning the score with neural networks using the current set of particles, using an objective derived from Stein's identity. Two synthetic experiments are done to demonstrate the effectiveness of the proposed method.


**Summary Of The Review:**

Given the lack of motivation for the proposed method and the somewhat misleading theoretical development, I am leaning toward rejection of this paper.

---

> ### Author Response · Authors · 2022-11-18
> **Reply to Reviewer LTVr**
>
> We thank the reviewer for their care in considering our submission, as well as their interest in the method. Below, we address their main concerns.
>
> **Motivation for learning the score.** We thank the reviewer for bringing up this point of confusion, which we have made more clear in the revised version. We would first like to make explicit that the score is *not* the only new quantity gained: the ability to compute $\rho_t(\cdot)$, and most importantly for our paper, the entropy production rate $\tfrac{d}{dt}H_t$, are also available. These *cannot* be computed from the SDE without doing an additional post-processing step, such as learning a flow based on SDE samples. We address this point next.
>
> **Post-processing the SDE.** We show in the original submission that $D_{\mathsf{KL}}(\rho_t  | \rho_t^*)$ is controlled by our proposed self-consistent procedure. As discussed in the common reply, in the revised version, we also show that neither $D_{\mathsf{KL}}(\rho_t | \rho_t^*)$ nor $D_{\mathsf{KL}}(\rho_t^* | \rho_t)$ are controlled by training on the SDE data. We further demonstrate empirically that estimates of the moments and/or the entropy are incorrect when training from the SDE, which highlights the need for our self-consistent approach. This form of training -- i.e., learning $s_t$ on the SDE samples -- is the most natural way to perform density estimation based on SDE samples alone, as it fundamentally exploits the structure of the problem to build the transport map.
>
> **Computation of the entropy and $\rho_t(X_t(x))$.** The reviewer is incorrect in thinking that these quantities require the entire history of the particles. We show in the text that $\tfrac{d}{dt}H_t = \mathbb{E}_{\rho_t}[\nabla\cdot v_t]$ and $\tfrac{d}{dt}\log\rho_t(X_t(x)) = -\nabla\cdot v_t(X_t(x))$. These equations show that the estimates can be computed incrementally throughout the simulation, and hence the particle trajectories and past parameters can be discarded.
>
> **(Old) Proposition 1 out-of-place.** As addressed in the common reply, we have updated our presentation of the theory to handle this point of confusion, and we thank the reviewer for suggesting to do so. In the new version, we show how Proposition 1 can be obtained systematically from a bound on $\frac{d}{dt}D_{\mathsf{KL}}(\rho_t | \rho_t^*)$. We then show how sequential SBTM can be obtained by directly minimizing this bound on the derivative, which is a computationally efficient relaxation of the constrained optimization problem. A-posteriori, control on these derivatives may be integrated to obtain control on the KL divergence at any $t$, and the output of the sequential procedure is feasible for the original problem; this provides the connection between our approach and our theoretical development. To address the reviewer's concern, we have discarded the use of $\lambda(t)$ altogether.
>
> **What is $G_t(x)$?** We thank the reviewer for addressing this point, because it is somewhat tricky to keep track of the different variables. $G_t(x)$ is *not* "the true score". There are essentially three scores in our work:
> 1. $s_t(x)$, the learned, approximate score.
> 2. $\nabla\log\rho_t(x)$, the score of the push-forward distribution $\rho_t = X_t\sharp \rho_0$.
> 3. $\nabla\log\rho_t^*(x)$, the score of the target solution $\rho_t^*$ that solves the Fokker-Planck equation.
>
> Our ultimate goal is to make $s_t(x) = \nabla\log\rho_t(x) = \nabla\log\rho_t^*(x)$. The quantity $G_t(x) = \nabla\log\rho_t(X_t(x))$, i.e, it is the score of the push-forward evaluated along particle trajectories. It is important to note that $s_t(X_t(x)) \neq G_t(x)$ in general -- this is exactly what the SBTM loss and sequential SBTM seek to promote. Remarkably, our theory shows that enforcing this consistency systematically improves the KL divergence $D_{\mathsf{KL}}(\rho_t | \rho_t^*)$, despite the fact that $\nabla\log\rho_t^*$ is never accessed (even implicitly).
>
> To make this point more clear, we have carefully modified the text so that all solutions of the Fokker-Planck equation are denoted by $\rho_t^*$. Originally, we presented the Fokker-Planck equation using the notation $\rho_t$, which may have caused some confusion.
>
> **Usefulness of $\tfrac{d}{dt}H_t$ and non-triviality of the systems** As explained in the common reply, the entropy production rate is a quantity of interest as it captures nonequilibrium aspects of the dynamics. For example, in the context of the interacting particle systems treated in the paper, the entropy production rate measures how much work is done by the trap on the particles and vice-versa. We would also like to stress that, while perhaps academic, these examples are not trivial, as they are high-dimensional (standard PDE integration methods are not available) and they allow us to demonstrate that our method can compute quantities not easily accessible otherwise.

---

> > ### Comment · Reviewer_LTVr · 2022-12-08
> > **Thanks for the reply and remaining concerns**
> >
> > Sorry for the late response. I appreciate the authors' reply and their effort in revising the text. The clarity of the revised paper has greatly improved. I appreciate the additional comparison with SBDM and the explanation for computing the entropy and the density incrementally.
> >
> > I remain concerned about the following aspects of the work. As these aspects are crucial, I would like to keep my current score unchanged.
> >
> > - Proposition 2 (the one that involves $G_t$) still seems out of place to me. The result is somewhat interesting on its own but it seems to have very little to do with the rest of the paper. I don't consider SSBTM as a relaxation of SBTM. For one, the ODE of $G_t$ does not correspond to anything in the formulation of SSBTM. And as reviewer qgFR also pointed out, the integration-by-part trick that leads to (SSBTM) is the same as the standard score-matching trick (e.g. (2) in https://arxiv.org/pdf/1905.07088.pdf is identical to (SSBTM) with $D_t$ being the identity matrix).
> > - The overall algorithm of the present paper is very similar to [1]. I think the only difference is in [1] they learn the vector field that approximates the Wasserstein gradient of KL, while the present paper learns the score, but they only differ by the drift (if assuming diffusion $D_t$ is identity). Granted the present paper can incorporate non-gradient flows but the extension is immediate. Both papers simulate particles driven by learnable vector fields.
> > - All experiments are done in 2D, which is very low (despite the authors claiming the total dimensionality of the PDE is large). Even the number of particles is very small (at most 50). The new animations are cool to watch, but I still consider the experimental contribution of the present paper quite weak. Comparison with recent JKO-based gradient flow methods is entirely missing (for instance [2][3][4] where they consider ambient dimension >50, and even without spatial discretization). Comparison with [1] may also be considered. At the very least, quantitative experiments with the Ornstein-Uhlenbeck process similar to the ones in [2-4] in varying dimensions should be considered.
> >
> >
> > [1] di Langosco, L. L., Fortuin, V., & Strathmann, H. (2021). Neural variational gradient descent. arXiv preprint arXiv:2107.10731.
> >
> > [2] Alvarez-Melis, D., Schiff, Y., & Mroueh, Y. (2021). Optimizing functionals on the space of probabilities with input convex neural networks. arXiv preprint arXiv:2106.00774.
> >
> > [3] Mokrov, P., Korotin, A., Li, L., Genevay, A., Solomon, J. M., & Burnaev, E. (2021). Large-scale wasserstein gradient flows. Advances in Neural Information Processing Systems, 34, 15243-15256.
> >
> > [4] Fan, J., Taghvaei, A., & Chen, Y. (2021). Variational Wasserstein gradient flow. arXiv preprint arXiv:2112.02424.

---

> > > ### Author Response · Authors · 2022-12-08
> > > **Additional mathematical detail**
> > >
> > > We now explain why the methods proposed in references [1], [2], [3], and [4], while interesting and relevant to our work, do not lend themselves to a comparison with our method. In essence, they all seek to solve a different problem.
> > >
> > > If the target probability density $\rho_t^*$ solves a continuity equation
> > >
> > > $$\partial_t \rho_t^* + \nabla \cdot j_t^* = 0$$
> > >
> > > for some probability current $j_t^*$, under mild conditions we can always find a potential $U_t^*$ such that $\rho_t^*$ also solves $\partial_t \rho_t^* + \nabla \cdot (\nabla U_t^* \rho_t^* ) = 0.$ Indeed, it suffices to obtain $U_t^*$ by solving the Poisson equation
> > >
> > > $$\Delta U_t^* = \nabla \cdot j_t^*,$$
> > >
> > > which admits a solution by the Fredholm alternative because the right-hand side integrates to zero. This is the Helmholtz decomposition, and it also relates to the Brenier polar decomposition.
> > >
> > > The equation
> > >
> > > $$\partial_t \rho_t^* + \nabla \cdot (\nabla U_t^* \rho_t^* ) = 0$$
> > >
> > > is a gradient flow in a Wasserstein metric that can be solved via the JKO procedure, which is the aim of references [2], [3], and [4]. However, it is important to stress that in general
> > >
> > > $$\nabla U_t^* \rho_t^* \not= j_t^*.$$
> > >
> > > That is, **this decomposition fails to capture any part of the probability current that does not affect $\rho_t^*$.** For example, for a non-equilibrium steady-state density $\rho^*$ (i.e. a steady state solution of the continuity equation above), then $U^* = 0$ but $j^* \neq 0$ in general. This is apparent in the active swimmer example treated in the appendix of our paper, which has a non-equilibrium invariant density and a non-zero invariant probability current. Our method is aimed at capturing both $\rho_t^*$ and $j_t^*$, whereas Refs. [1]-[4] only care about $\rho_t^*$ or $\lim_{t\rightarrow\infty} \rho_t^*$. Estimating the probability current is physically relevant in many applications. For example, the entropy production rate *density*, a quantity of interest in the active matter community, is $j_t^*(x) \cdot \nabla\rho_t^*(x) \neq \nabla U_t^*(x) \cdot \nabla\rho_t^*(x)$.

---

> > > ### Author Response · Authors · 2022-12-08
> > > **Response to additional concerns**
> > >
> > > We thank the reviewer for these additional comments. We are glad that they appreciate the clarity of the new version of our paper, as well as the recent revisions. Let us address their remaining three concerns:
> > >
> > > 1.  Proposition 2 follows naturally from the bound on the entropy in Eq. (9) (which is itself a direct consequence of Proposition 1), and is used to highlight a key aspect of SBTM that distinguishes it from SBDM. Namely, ***SBTM is self-contained and does not require any input from the density $\rho_t^*$ that solves the FPE.*** We believe that this is important to stress in the main text before discussing SSBTM. Moreover, Proposition 2 underlines a second key aspect of the method concerning Proposition 3. While the integration by parts is formally identical to the standard approach in the score matching literature, it is both conceptually and practically different because it is based on the current estimate $\rho_t$ -- given by the solution to the transport equation $\partial_t \rho_t = -\nabla\cdot (v_t\rho_t)$ -- and *not* $\rho^*_t$, which solves the FPE. That is, no external data from the original SDE is ever needed.
> > > 2.   Our method does present similarities with the references [1], [2], [3], and [4] suggested by the reviewer, but ***the overall goal of these works is very different from ours.*** Refs. [1], [2], [3], and [4] all aim to sample from a (fixed) target density. The schemes proceed by either estimating a Wasserstein gradient (Ref. [1], extending Stein variational gradient descent), or via the JKO scheme (Refs. [2]-[4]). By contrast, we aim to compute a time-dependent target $\rho^*_t$ that solves an FPE using only structural information about the drift and diffusion that enter the equation; we also aim to estimate the probability current associated with this dynamics. These goals add the additional complexity that the non-equilibrium sample dynamics must be captured correctly at all times. In particular, ***this dynamics does not reduce to a gradient flow in the Wasserstein-2 metric. More specifically, the velocity $v_t$ that we seek is not the gradient of a potential in general.***
> > > 3.   ***Our experiments are not performed in 2D, but are high-dimensional!*** They involve systems of particles that each evolve in 2D, but interact amongst themselves in a non-trivial way so that the overall dimensionality is $d=2N$ if there are $N$ particles, not $d=2$. With $N=50$, this results in a system in $d=100$ dimensions, i.e. ***the resulting FPE is higher-dimensional than nearly all of the numerical example in Refs. [1], [2], [3], and [4].*** In particular, our first example with 50 particles *is* an Ornstein-Uhlenbeck (OU) process, and it is higher-dimensional than the OU examples treated in Refs. [1], [2], and [4]. In addition, our example has the added complexity that the drift coefficient is time-dependent, so that there is no equilibrium density, i.e., $\lim_{t\rightarrow\infty} \rho_t^*$ does not exist. Our second example with $N=5$ has dimensionality $d=10$, which is already high for discretization-based solvers and is similar to the dimensionality in many of the examples in Refs. [1], [2], [3], and [4]. This example was chosen to highlight the complexity of the dynamics captured by the probability flow -- we can scale this second example to even higher dimension, but the flow is less easily interpretable visually. Indeed, we have studied the same system with $N=50$ particles, but found the dynamics more interesting with $N=5$ particles. In any case, this concern regarding dimensionality is not valid.
> > >
> > >
> > > ### Summary of differences from references
> > > **Ref. [1]** studies an extension of Stein variational gradient descent, with a goal to sample from a fixed Bayesian posterior. The goal is not to resolve temporal dynamics as in our setting. The examples are all of comparable dimension to ours, except for their experiments on MNIST.
> > >
> > > **Ref. [2]** studies a JKO-based scheme for simulating Wasserstein gradient flows arising from problems of the form $\min_\rho F[\rho]$, a class of problems in which our examples do not fit. As discussed above, a JKO scheme will not accurately represent the probability current in our examples. The numerical examples studied are primarily low-dimensional PDEs (of lower dimension than our OU system, for example).
> > >
> > > **Ref. [3]** is similar to Ref. [2], and studies a JKO scheme for sampling at equilibrium or for computing an equilibrium density. The techniques do not extend to the time-dependent dynamics considered here. The OU processes studied only go up to dimension $d=12$, an order of magnitude lower than ours; moreover, they are simpler because they are not time-dependent.
> > >
> > > **Ref. [4]** uses similar ideas to Refs. [2]-[4], making use of a variational representation of the $f$-divergences appearing in the objective $F[\rho]$. Except for their image experiments, the dimensionality of the numerical examples is comparable to or lower than ours.

---

### Official Review · Reviewer_qgFR · 2022-10-24

**Confidence:** 5
**Correctness:** 3
**Technical Novelty And Significance:** 3
**Empirical Novelty And Significance:** 3
**Recommendation:** 6

**Clarity, Quality, Novelty And Reproducibility:**

This paper is well-written. There is an overlap with some recent work, which is acknowledged by the authors.

**Strength And Weaknesses:**

Strength: While the score-based transport model is similar to a recent work (Shen et al. 2022), as acknowledged by the authors, the authors generalizes the previous art by considering a non-constant diffusion coefficient and show that the objective of SBTM controls the KL divergence between the hypothesis density and the ground-truth density, which improves over the previous Wasserstein-type bound.

Weakness:
1. Algorithm 1 can be memory consuming since it requires a individual neural network model for the score function at every time-stamp. Please correct me if I am wrong.
2. I do not see how Proposition 3 differs from the standard score matching technique. This seems the exact score-matching technique for approximating $\nabla \log \rho_t$.
3. It is not clear how the optimization problem (SBTM) is solved and the analysis on how Algorithm 1 solves SBTM, e.g. the convergence guarantee, is missing.
4. Since SBTM is proposed as an improvement over SBDM, it seems reasonable to include SBDM in the empirical study, but I do not seem the comparison between SBTM and SBDM.



**Summary Of The Paper:**

This paper considers the transport-map approach to learn the score function along the solution trajectory of the Fokker-Planck equation. In contrast to the score-matching approach, the proposed proposed sequential SBTM method does not require to simulate the underlying SDE. Instead, the particles are propagated via a sequentially constructed estimation of the score function. Experiments are conducted to show the empirical advantage of the proposed method.

**Summary Of The Review:**

Overall I think the self-consistency based approach is a promising direction for score estimation and this paper makes a good contribution.

---

> ### Author Response · Authors · 2022-11-18
> **Reply to Reviewer qgFR**
>
> We thank the reviewer for their thoughtful suggestions to improve the work, as well as  their careful evaluation of the manuscript. Below, we address their main criticisms.
>
> **Memory requirements.** It is true that the sequential SBTM scheme requires a new set of network parameters at each time $t$. However, we find that small networks are sufficient for the tasks we consider, so that the resulting memory requirement is not very significant. To eliminate this additional demand entirely, all desired outputs (evaluation of $\rho_t$, estimate of $\tfrac{d}{dt}H_t$ or $H_t$, moments of the samples) can be computed on-the-fly, and all prior parameters can be discarded. Moreover, the optimization need only produce $\theta_{t+\Delta t}$ with $\theta_t$ known, so very little optimization is needed at each timestep.
>
> **Difference with standard score matching.** As explained in the `Common Reply', the main difference is that SBTM learns the score self-consistently using the solution from the probability flow ODE itself rather than external data from the SDE. In particular this allows us to estimate $\nabla \log \rho_t(x)$ along this flow, i.e. we have access to $G_t(x) = \nabla \log \rho_t(X_t(x))$, which is not a quantity available in standard diffusion-based score matching.
>
> **Comparison to SBDM.** The problem setting of SBDM (generative modeling) is different than what we consider here, so there is no direct comparison between SBTM and SBDM. However, a natural comparison would be to train the transport map on samples from $\rho_t^*$ produced via simulation of the SDE, then use this transport map to calculate quantities that are not directly accessible from the SDE, such as the entropy production rate. As discussed in the common reply, we implemented this in the updated version. We show that the scheme is significantly less stable than using our sequential SBTM formulation, and that theoretically neither $D_{\mathsf{KL}}(\rho_t | \rho_t^*)$ *nor* $D_{\mathsf{KL}}(\rho_t^* | \rho_t)$ are controlled when training on SDE data.

---

### Official Review · Reviewer_1NXr · 2022-10-24

**Confidence:** 4
**Correctness:** 4
**Technical Novelty And Significance:** 3
**Empirical Novelty And Significance:** Not applicable
**Recommendation:** 6

**Clarity, Quality, Novelty And Reproducibility:**

## Clarity
The text is clearly written, easy to follow, with appropriate references to the literature.

## Quality
I think that quite a bit of work should be done to do comparison with reasonable baselines (i.e. take compute time into account and compare to density-estimation procedures exploiting the same problem-structure)

## Novelty
Except the related work of (Maoutsa et al) that exploit similar ideas but in lowish dimension, the proposed methodology appears to be new.

## Reproducibility
all good

**Strength And Weaknesses:**

**Strength:**
The method is very intuitive and appears to work well on simulated examples.

**weakness:**
1. the proposed scheme does not seem to very stable in the sense that errors in estimating the score may amplify as the particles are propagated. In some sense, the vanilla approach consisting in simulating forward the associated SDE appears to be much more stable in that respect. Is this true? Can the authors discuss and possibly illustrate this empirically.

2. I think that the simulation should take compute-time into account. The vanilla Monte-Carlo approach (ie. simulate forward the SDE) is extremely scalable. The proposed approach requires implementing score-matching at each step.

3. Evaluating $\rho_t$ from samples is basically a density estimation problem. I think the authors should consequently consider comparing their method to Vanilla-Monte-Carlo + density estimation (eg. normalising flow, or something of that sort). Indeed, there are also quite a few ways to estimate differential entropies, and mutual information, from samples!

4. The network architectures are exploiting quite a lot the structure of the problem. One may argue that this structure could also be used to estimate the density out of samples much more efficiently than a completely problem-agnostic approach ...


**minor question**:
1. is it $-\alpha$ in Equation (11) or $+\alpha$

**Summary Of The Paper:**

In high dimensional setting, it is typically impossible to approximate the solution $\rho_t$ to a Fokker Planck Equation (FPE) with conventional grid-based methods. A  standard method consists in considering the associated SDE a simulate many trajectories from this SDE to collect statistics of the solution at any future time $t$. This methods allows to compute any moment $\int \mathcal{O}(x) \rho_t(dx)$ of the solution (eg. Monte-Carlo method), but the evaluation of $\rho_t(x)$ is typically not available with this basic approach.

The article proposes to express the solution of the FPE as a transport equation $\partial_t \rho_t = -\nabla \cdot (v_t \rho_t)$ with a velocity field $v_t$ that is expressed as a function of the score $s_t$ of $\rho_t$ itself, $s_t = \nabla \rho_t$. This motivates the following particle approach:

1. start from $N$ samples $x_1, \ldots, x_N$ samples from $\rho_0$
2. use (some variant of) score matching to evaluate the score of $\rho_t$ and construct the velocity field $v_t$
3. push forward the particles through the transport equation to get a particle approximation of $\rho_{t + \Delta t}$
4. iterate: go back to 2

The advantage of this approach is that, because $\rho_t$ is expressed as transport equation, is is straightforward to evaluate $\rho_t$ along trajectories of the transported particles (contrarily to the SDE approach).

**Summary Of The Review:**

The proposed method is natural and very interesting. I would like to encourage the authors to work on their simulations in order to more convincingly demonstrate the appeal of the approach when compared to more standard methodologies.

---

> ### Author Response · Authors · 2022-11-18
> **Reply to Reviewer 1NXr**
>
> We would like to thank the reviewer for their careful reading, positive view on the manuscript, and constructive suggestions. Below, we address the main points raised.
>
> **Errors in the score.** The reviewer is correct to note that errors in the learned score can in principle amplify with time in the sequential SBTM scheme. While this could be a problem for some  systems, we did not find this to be a concern empirically in the examples that we study. Worst-case amplification of errors is standard in the numerical analysis of timestepping methods; in fact, *all* ODE and SDE discretization schemes have error bounds that scale exponentially in time. For SDE integration schemes such as the Euler-Maruyama method often applied in practice, this applies to both the weak and strong errors. Despite these bounds, timestepping schemes work very well for most systems and form the backbone of scientific computing applications that involve ODEs or PDEs. Hence, as long as the optimization tolerance is consistent with the order of accuracy of the integrator for the probability flow equation, learning the score will not lead to more accumulation of error than standard timestepping methods.
>
> It is worth noting that accumulation of errors could be eliminated by using the original (time-integrated) variant of the SBTM loss. However, we opt for the sequential approach in this work to avoid the computational expense of the full SBTM loss. In accordance with the reviewer's concerns, we have added a short paragraph discussing these points after the introduction of the sequential method.
>
> **Compute time.** The reviewer is correct that the SDE is computationally cheaper. The price paid for this reduction in complexity is an inability to evaluate $\rho_t$ or to compute the entropy $H_t$. Nevertheless, in practice, we find that sequential SBTM is fairly cheap: for example, for a $10$-dimensional Fokker-Planck equation, $n=10^4$ samples, a horizon of $T = 10$, and a timestep of $\Delta t = 10^{-3}$, we find that sequential SBTM only takes around two hours on a single NVIDIA RTX8000 GPU. We have added time scaling information in the main text emphasizing this point.
>
> **Comparison to density estimation.** We agree with the reviewer that this is an important comparison to make, and we have done so in the revision. Please see the common reply for details: we have added both theoretical and empirical support that such a procedure will be less effective than sequential SBTM.
>
> **Minor points.** It is $+\alpha$, we have fixed this, thank you.

---

> > ### Comment · Reviewer_1NXr · 2022-11-21
> > **follow-up comments**
> >
> > Thank you very much for your thorough reply:
> >
> > **Errors in the score:** sorry for being unclear. I am referring to the fact that because you are propagating through an ODE whose velocity is learned from samples, this can potentially significantly add more instabilities on top of the "standard" instabilities associated to time-stepping methods. In other words, the vanilla Monte-Carlo method indeed also suffers from some instabilities (eg. Euler-Maruyama / etc..) but the proposed method seems to suffer from an **additional** layer of instability. It may be worth just adding a few comments in the text.
> >
> > **Compute time:** I think that when comparing different methods, with different tuning parameters/etc.., one would expect a figure with compute on the x-axis and some measure of accuracy on the y-axis. I do not find the statement "we find that sequential SBTM is fairly cheap" entirely satisfying. It would be best to compare it to a baseline (eg. Vanilla Monte-Carlo, or Vanilla Monte-Carlo with density estimation) instead of providing absolute numbers.
> >
> > **Comparison to density estimation:** thank you for the additional simulations. Could you please point out precisely where the simulations are when Vanilla MC run for roughly the same compute time is compared to the proposed method, and demonstrating that the moment estimate are less accurate?
> >
> > **Computing entropy:** there are quite a few methods for computing entropies from samples (eg. literature review of [1])
> >
> >
> > [1] Pichler, G., Colombo, P. J. A., Boudiaf, M., Koliander, G., & Piantanida, P. (2022, June). A differential entropy estimator for training neural networks. In International Conference on Machine Learning (pp. 17691-17715). PMLR.

---

> > > ### Author Response · Authors · 2022-11-22
> > > **Response to the follow-up comments**
> > >
> > > **Errors in the score.** We thank the reviewer for brining this point of confusion to our attention, and we will add a few additional comments about accumulation of error in the text. On this topic, we would also like to stress the following:  as long as the error of the learned score is consistent with the error of the ODE integrator *on the training samples*, there will be no additional accumulation of error relative to standard timestepping methods. This is because SSBTM only uses the learned score on samples, so that strictly speaking, the generalization performance of the model does not affect its ability to propagate accurately. We believe that this is primarily why training self-consistently with SSBTM results in greater accuracy than SBDM. Indeed, optimization for $s_t$ at each $t$ is simply a method to timestep the score, rather than timestepping each gridpoint as in standard PDE solving methods. Finally, let us emphasize that error accumulation can in principle be avoided completely by using the SBTM time-integral loss.
> > >
> > > **Compute time.** We chose to report the timing for our experiments in the main text, rather than explicitly compare to other methods because it is rather challenging to make such a comparison in a manner that is scientifically sound and fair. For example: if one computation is cheaper, but it requires running many sweeps over hyper-parameters to find a working model, should all of those sweeps be counted in the total time?  Moreover, the other methods proposed for comparison (barring density estimation from SDE samples discussed below) do not give access to all of the same quantities as SBTM, so that a timing comparison alone may be misleading.
> > >
> > > **Comparison to density estimation.** To facilitate a fair comparison that does not require different model classes, we performed the optimization over samples from the SDE sequentially, i.e. by minimizing $\frac1n \sum_{i=1}^n\big(|s_{\theta_t}(x_i^*(t))|^2 + 2\nabla\cdot s_{\theta_t}(x_i^*(t))\big)$ with $\{x_i^*(t)\}_{i=1}^n$ independent trajectories from the SDE. This was done with an identical model and an identical optimization procedure. Hence, the computational cost is identical for both methods, as timestepping the SDE is no cheaper than timestepping the ODE. As explained above (and demonstrated in text), the probability flow ODE that uses the score learned this way gave much less accurate results than the one learned self-consistently via SSBTM.
> > >
> > > In principle, a model $s_\theta(t, x)$ could be used to minimize $\int_0^T \frac1n \sum_{i=1}^n\big(|s_\theta(t, x_i^*(t))|^2 + 2\nabla\cdot s_\theta(t, x_i^*(t))\big)$ globally, rather than using a sequential procedure, so that the task of learning of the score is independent of the time-discretization grid used to integrate the probability flow ODE. However, if this approach is used, it becomes challenging to ensure that the two models ($s_{\theta_t}^{\text{SBTM}}(x)$ and $s_\theta^{\text{SBDM}}(t, x)$) are on the same footing, and hence to isolate the source of improved performance of one method. We also do not expect that training in this way would improve the results significantly. It is for these reasons that we chose to perform the optimization sequentially.
> > >
> > > **Computing the entropy from SDE data.** We thank the reviewer for bringing this reference to our attention: we will cite it in the camera-ready version. Most methods for entropy estimation, such as those covered in the literature review of the suggested reference or the $\texttt{KNIFE}$ method of the reference itself, amount to a form of kernel density estimation (KDE) and use $H_t \approx -\frac{1}{n}\sum_{i=1}^n \log\hat{\rho}_t^{\text{KDE}}(x_i)$.
> > >
> > > For the re-submission, we performed a comparison to kernel density estimation with a fixed bandwidth using SDE data (no score learning). This approach gave a very bad prediction that was comparable to the prediction of the noise-free system. For this reason, we found the comparison to training on the SDE data more compelling, and chose only to include those results. In the camera-ready version, we can include the KDE comparison in addition.
> > >
> > > The $\texttt{KNIFE}$ method could give better results than basic KDE, but its computational overhead is more significant than standard KDE, and the vanilla method would likely need to be amended to our setting. Indeed, our approach allows us to easily incorporate permutation symmetry in the learning by using an equivariant ansatz for the score, which is key for scalability. Permutation symmetry could in principle be accounted for with KDE methods such as $\texttt{KNIFE}$, but we feel that exploring this direction is beyond the scope of the present paper.

---

> > > > ### Comment · Reviewer_1NXr · 2022-11-23
> > > > **Thank you.**
> > > >
> > > > Thank you for providing these additional details.

---

> > > > > ### Author Response · Authors · 2022-12-06
> > > > > **Thanks to you too.**
> > > > >
> > > > > Thanks to you as well for providing feedback and for engaging in a dialogue with us. Please let us know if there is anything else we can do to convince you to increase your score any further.

---

### Author Response · Authors · 2022-11-18
**Common Reply (2/2)**

### Comparison to  SBDM
The reviewers have asked how SBTM compares to SBDM, i.e.  what happens if we estimate a transport map of the form in the main text -- $\dot{X}_t(x) = b_t(X_t(x)) - D_t(X_t(x))s_t(X_t(x))$ -- with $s_t$ learned directly on data from the SDE. This forms a natural comparison for our method, as it preserves the structural aspects of the problem that we exploit for SBTM and only changes the training data from self-consistent to external. However, we show that this approach is less accurate than SBTM both in theory and in practice:

**Theoretically:** In Appendix B.5 of the updated version, we show that neither $D_{KL}(\rho_t | \rho_t^*)$ *nor* $D_{KL}(\rho_t^* | \rho_t)$ are controlled when training the score on data from the SDE. This is a subtle point: previous work in the context of SBDM [2] shows that training on SDE data leads to a bound on $D_{KL}(\rho_t^* | \rho_t^{SDE})$ where $\rho_t^{SDE}$ is the density of the reverse-time SDE used to sample new images. This is *not* true when $\rho_t$ is the density of the ODE, which is necessary in our work for computation of the entropy production rate $\tfrac{d}{dt}H_t$ and for evaluation of $\rho_t$.

**Experimentally:** In the revision, we demonstrate empirically that learning on the SDE samples is numerically less stable than the self-consistent sequential SBTM procedure, consistent with our theory. For many hyper-parameter choices, training on the SDE data leads to highly inaccurate moments, and we found that it always led to a divergent prediction of the entropy. In addition to our theory, there is an intuitive explanation for this observation. Learning on the SDE samples requires generalization to areas of state space probed by the ODE, which may be different than those probed by the SDE. When training on ODE data, generalization is less crucial -- all that is needed is good performance on the training set.


### Comparison to other models such as normalizing flows
There are many ways one could try to perform a comparison between sequential SBTM and alternative methods for density estimation on $\rho_t^*$, which makes it difficult to ensure that the two approaches are on an even playing field. An option such as a normalizing flow, as suggested by Reviewer 1, would require training a *separate* flow to map $\mathsf{N}(0, 1)$ to $\rho_t$ for each $t$. This would inevitably be more expensive than the sequential SBTM approach, which only needs to map $\rho_{t}$ to $\rho_{t+\Delta t}$ at each $t$, a much simpler problem. For this reason, in the revision we focused on the comparison to SBDM, as the two methods are more comparable.

### References
[1] Cesare Nardini, Étienne Fodor, Elsen Tjhung, Frédéric van Wijland, Julien Tailleur, and Michael E. Cates. Entropy production in field theories without time-reversal symmetry: Quantifying the non-equilibrium character of active matter. Phys. Rev. X, 7:021007

[2] Yang Song, Conor Durkan, Iain Murray, and Stefano Ermon. Maximum Likelihood Training of Score-Based Diffusion Models. NeurIPS 2021.

---

### Author Response · Authors · 2022-11-18
**Common Reply (1/2)**

We would like to thank all of the reviewers for their careful reading, constructive suggestions, and overall interest in our work. In this reply, we collect the main suggestions common to all reviewers; below, we address their concerns individually. For ease of the reviewers, we have uploaded a new version of the text that tracks changes spread throughout the paper from the old version in red. In addition, as described below, Section 3 has been re-worked to address the comments of the reviewers. In the new version, we have also attached the appendix to the main pdf file.

### Clarifying the outputs of SBTM
To emphasize the motivation of SBTM, we have now better demonstrated that SBTM allows for the calculation of quantities that are not directly accessible from trajectories of the SDE. Our main target in this work is the entropy production rate (EPR), a quantity of strong interest in the active matter community as a measure of the non-equilibrium aspects of the dynamics [1]. To highlight this point, we have updated the example of five soft spheres to emphasize the prediction of the EPR, and we now consider both circular and linear motion of the trap, which lead to distinct behaviors of the EPR. Moreover, as discussed further below, we perform a comparison to show that a prediction of the EPR is not easily obtained from SDE samples.

We would also like to stress that, from a physical standoint, having the ability to visualize the flowlines of the probability current is informative in its own right. This is captured by the movies of the particle motion, which we have now linked to directly from the main text. We strongly encourage the reviewers to view the movies, as the still frames in the paper do not do the motion full justice. They are available [here](https://drive.google.com/drive/folders/1JS9H7f9G0JnMe8_Fin22bf6bl3Drvid9).


### Clarifying the relation between SBTM and Sequential SBTM
In the revision, we have streamlined the presentation of our theory. We feel that this new treatment makes the connection between sequential SBTM and the original SBTM loss significantly more clear. In particular, in the new version, we show that:
1. The SBTM  optimization procedure follows naturally from a bound on $\frac{d}{dt}D_{\mathsf{KL}}(\rho_t | \rho_t^*)$. This bound allows for control of $D_{\mathsf{KL}}(\rho_T | \rho_T^*)$ up to any $T\in[0,\infty)$ provided that we learn the flow in a *self-contained fashion* using walkers $X_t(x)$ propagated by the probability flow itself.
2. Sequential SBTM is an alternative to SBTM, whereby $\frac{d}{dt}D_{\mathsf{KL}}(\rho_t | \rho_t^*)$ is controlled at each time $t$ as a proxy to control $D_{\mathsf{KL}}(\rho_t | \rho_t^*)$. Intuitively, this is a relaxation of the constrained optimization problem that learns to prevent increase in the error. This avoids backpropagation through ODE solves, as well as the need to solve the equation for $G_t(x)=\nabla \log \rho_t(X_t(x))$, which together render the non-sequential SBTM computationally expensive.
3. Sequential SBTM ensures control on $D_{\mathsf{KL}}(\rho_t | \rho_t^*)$ via integration *a-posteriori*, and it produces a feasible solution for the original SBTM problem. Hence, it solves the original problem while avoiding direct gradient-based optimization on the constrained SBTM loss.

---

### Author Response · Authors · 2022-12-06
**Request for further comments**

Dear Reviewers,

Thanks again for your helpful questions and comments about our work.  Following your suggestions, we have better explained how our approach differs from score-based diffusion modeling,  highlighted its advantages, clarified the outputs, and expanded both the experiments and theory.

Any further feedback on our rebuttal or the revision would be greatly appreciated, and we would gladly address any remaining concerns you may have about our work.  As a reminder, the summary of how we addressed your initial reviews can be found in the two-part common reply, with more details given in the individual replies to each of you.

All the best.

---

### Decision · Program_Chairs · 2023-01-20

**Decision:**

Reject

**Justification For Why Not Higher Score:**

All the reviewers are leaning toward reject after the AC-reviewer meeting.

**Justification For Why Not Lower Score:**

N/A

**Metareview: Summary, Strengths And Weaknesses:**

This paper presents a learning based approach for solving the Fokker-Planck equation. Rather than generating samples via integration of the associated stochastic differential equations, this submission considers a transport-map approach, updating particles via a sequentially constructed estimation of the score function. On the positive side, the proposed method builds a nice connection between the KL divergence of the estimated probability flow and the score matching objective. Overall, the paper is written clearly and the presentation is pleasant. However, the reviewers have raised questions regarding the novelty of the proposed method, and the lack of thorough comparisons against relevant baselines and competing methods. Given these concerns, the reviewers and AC do not find the paper ready for publication at ICLR.

**Summary Of Ac-Reviewer Meeting:**

2 out of 3 reviewers including qgFR and LTVr joined the meeting. The 3rd reviewer shared their feedback via email with AC.

Both reviewers present in the call have worked on similar topics.

LTVr rated the paper borderline reject mentioning:
- The novelty of this submission is limited due to previous works such as Neural Variational Gradient Descent
- The experiments are conducted in a low dimensional space (often particles in 2D), even though prior works such as JKO decomposition have been tested on much larger spaces
- The reviewer acknowledges that proposition 2 is an interesting result of this submission but they emphasize that the paper actually does not use this proposition directly
- They question whether it is easy to show that proposition 3 is a relaxation of proposition 2

qgFR agrees that the issues raised by LTVr are not addressed properly in the rebuttal. They also agree that JKO baseline is a relevant baseline that should be included in comparisons. On the positive side, they acknowledge that the connection between SBTM to the KL divergence comes with nice guarantees for training.

The third reviewer 1NXr would expect more exhaustive comparisons against Monte Carlo-based approaches that scale easily in high-dimensional spaces, do not require training a neural network per iteration, and are potentially even faster than the proposed method. At the moment, they are not finding this discussion very well implemented, and the rebuttal was not entirely clear to them.

---

> ### Author Response · Authors · 2023-02-06
> **Response to paper decision**
>
> Dear Area Chair,
>
> We appreciate your response, and we accept the decision of the reviewers to reject our paper. However, we are concerned that several statements made in the reviews, as well as the final meta-review, are factually incorrect. Here, we hope to dispel this misunderstanding one more time:
>
> 1. Methods such as Neural Variational Gradient Descent, as explained in our replies to the reviewers below, solve a different problem than the one we consider here. Such techniques aim at **sampling from the equilibrium distribution** of a Fokker-Planck equation, while here we are interested in **resolving the temporal dynamics** of a Fokker-Planck equation. Indeed, most of our numerical examples **do not have an equilibrium distribution**, rendering such techniques **inapplicable**.
>
> 2. As stated repeatedly in our rebuttal to the reviewers and also in the text of the paper itself, the experiments are **not low dimensional**. The ambient space is $d=2$, and the number of physical interacting particles (not samples) varies from $N=5$ to $N=50$. The dimension of the Fokker-Planck equation is $N\times d$, ranging from $10$ to $100$, which is intractable for standard PDE solvers. Because we are interested in developing simulation techniques for problems in statistical physics, **high-dimensionality naturally arises in this way, as the motion of many interacting particles in a space of physical dimension two or three**. Again, we clarify that the number of physical interacting particles is completely distinct from the number of samples used to perform the learning.
>
> 3. Monte-Carlo methods are indeed fast, but they only allow one to compute expectations over the solution of the SDE, and unlike our proposed approach, they do not give direct information about the solution of the FPE. Since we are interested in computing physically interesting quantities such as the entropy, Monte Carlo methods are not applicable directly and would have to be combined with other approaches such as kernel density estimation, which do not scale well with dimension.  **In particular, we showed that learning from samples of the SDE gives worse performance than the proposed method.**
>
> Sincerely,
>
> The Authors